# Over expression of modified *Isomaltulose Synthase Gene II* (*ImSyGII*) under single and double promoters drive unprecedented sugar contents in sugarcane

Mudassar Fareed Awan[1]*, Sajed Ali[1], Muhammad Farhan Sarwar[1], Muhammad Shafiq[1], Usman Arif[1], Qurban Ali[2]*, Abdul Munim Farooq[3], Shiming Han[4]*, Idrees Ahmad Nasir[3]

1 Department of Biotechnology, Knowledge Unit of Science, University of Management and Technology, Sialkot Campus, Punjab, Pakistan, 2 Department of Plant Breeding and Genetics, University of the Punjab, Lahore, Pakistan, 3 Centre of Excellence in Molecular Biology, University of the Punjab, Lahore, Pakistan, 4 School of Biological Sciences and Technology, Liupanshui Normal University, Liupanshui, China

* mudassar.fareed@skt.umt.edu.pk (MFA); saim1692@gmail.com (QA); hanshiliang888@163.com (SH)

**Data Availability Statement:** All relevant data are within the manuscript and its Supporting Information files.

## Abstract

Sugarcane has been grown all around the world to meet sugar demands for industrial sector. The current sugar recovery percentage in sugarcane cultivars is dismally low which demands scientific efforts for improvements. Multiple approaches were adopted to enhance sugar contents in commercial sugarcane plants in contrast to conventional plant breeding methods. The exploitation of biotechnological methods and exploration of isomaltulose synthetic genes presented a promising solution to increase the existing low level of sugar recovery percentage in *Saccharum officinarum* L. *Isomaltulose synthase gene II* was employed and integrated into plant expression vector driven under the leaf and stem specific promoters terminated by nopaline synthase gene in a cloning strategy shown in the present study. Three gene constructs were developed in various combinations driven under promoters *Zea mays ubiquitin* and *Cestrum Yellow Leaf Curl virus* in the single and double combined stacked system. The transformation was executed in multiple formats with single transformed events, double promoter transformation events and triple construct stacked promoters in sugarcane induced calli via the particle gene gun. The transformation of *ImSyGII* in sugarcane genotype HSF-240 was confirmed by molecular gene analysis while expression quantification was determined through Real Time PCR. Furthermore, HPLC was also done to harvest the increased amounts of Isomaltulose in transgenic sugarcane juice. The present work upheld the enhanced *ImSyGII* expression in leaves owing to the exploitation of ubiquitin, while the Cestrum Yellow Leaf Curl virus promoter enhanced gene expression in sugarcane stems. The employment of three gene constructs collectively produced elite sugar lines producing more than 78% enhancements in whole sugar recovery percentage. The mature internode proved highly efficient and receptive regarding the production of isomaltulose. Quantifications and sugar contents evaluations upheld an increased Brix ratio of transgenic sugarcane lines than control lines.

**Funding:** This work was supported by the Key Laboratory of kiwifruit resources development and utilization of Guizhou Universities (Qian Jiaoji [2022] 054), projects of Liupanshui Normal University (Biological Science, LPSSYYlzy2003, LPSSY2023XKTD09, Lpssyzxxm202304, LPSSYKYJJ201601) and the Science and Technology project of Liupanshui City (Grant #52020-2020-0906,). The funder has provided financial help for study design, data collection and analysis, decision to publish, or preparation of the manuscript.

**Competing interests:** The authors have declared that no competing interests exist.

## 1.0. Introduction

Genetic engineering plays a pivotal role in the production of recombinant proteins. These recombinant proteins can be quantified through single cell as well as multicellular mechanisms. Many microbes, insects, mammals and transgenic animal systems are employed to yield proteins through genetic manipulations [1]. Different classes of plants and their living system have been utilized at large to produce high concentrations of recombinant proteins with multiple benefits [2, 3]. These benefits include enhanced production capacity, generation of wide varieties, high scalability, low input costs and eco-friendly [4]. Transgenic technology in sugarcane can be used to produce variety of biomolecules like Isomaltulose (IM) and Trehalulose (TH). The selection of appropriate hosts and expression vectors is very important because protein assembly and its accumulation directly depends upon these factors [5]. While selecting a purely plant-based expression factory, principal elements need to be considered which include vegetative production per hectare, protein recovery percentage, transformation efficiency, measuring ability and Biosafety [6].

Sugarcane is considered a major source of carbohydrates all around the world, producing nearly 75% of sugar as compared to other sugar crops. It is not only produces biofuels but also extensively used as a feedstock for animals. Sugarcane is a rapidly growing grass species possessing with well-systematic $C_4$ photosynthetic routes having the ability to enhance biomass production ability [4, 7]. It is considered very friendly to genetic manipulations with optimized transformation protocols and tissue culturing methods [8]. Being a vegetative propagated crop, sugarcane has very low rates of genetic pollution happened due to horizontal gene transfer [5, 9]. Many biomolecules were actively manufactured through sugarcane crops which was used as a bio-factory [10] for bioplastics, biofuels, sugar isomers and alternative carbohydrate molecules. The sugarcane stem constitutes a major fragment of total biomass which is the place for storage of high sugar contents in sugarcane stalk [11].

The primary purpose of sugarcane cultivation aims to produce increased amounts of sucrose (SUC) calculated as commercial cane sugar (CCS) in sugar Industry [4, 12]. The increased levels of diabetes day by day demand the production of such sugar from sugarcane which is more healthy and sweeter than SUC. Studies indicated that IM and TH are the SUC isomers found in honey and considered healthier than other SUC molecule. The IM and TH have less glycemic index with slower digestion rate causing very good effect in health as compared to SUC [13]. In addition to that the local sugarcane varieties manifest very less degree of sugar content [14, 15]. The deleterious effects from existing SUC produced by sugarcane demands an alternate source to SUC which may enhance sugar contents in local sugarcane cultivars and harvest healthful sweeteners [16]. These SUC isomers were produced by using SUC as a substrate with Sucrose Isomerase genes (SIGs) administration [17]. Studies indicated that IM and TH can replace SUC as sugar sources and can be synthesized by genetic manipulations of the sugarcane plants [18, 19]. Various types of sucrose isomerase genes encoded in different microbial species have been used to produce IM and TH in plants by genetic engineering. Similarly, Isomaltulose synthases (ImSy) are considered more valuable than other SUC isomerases because of their broad-spectrum applications in microbial community [20, 21]. Some studies also witnessed the 100% increase in overall sugar contents in sugarcane due to the induction of IM as bio-molecule [22]. The Isomaltulose synthases possess many novel properties which include the synthesis of IM, a highly beneficial sugarcane sweetening product [23].

The present study illustrated that sugarcane is a good candidate plant to be used as an expression system to cultivate and harvest high quantity of recombinant proteins. The *Isomaltulose Synthase Gene II (ImSyGII)* was codon optimized and synthesized according to the bacterial nucleotide sequence with accession no. AY223550 as reported in NCBI database. The synthetic

genes were synthesized from BioBasic Pvt and further used in our experimentation. Sugarcane crop has great potential to exhibit differential gene expression under single, double, and triple promoter stacked system. Different combinations of constitutive promoters including single and double with enhancer sequences, and stem specific targets were employed to calculate *ImSyGII* gene expression. Modified *ImSyGII* was cloned in plant expression vector, *pCAMBIA1301* and transformed in sugarcane through gene gun. The vacuolar targeted signal peptide was also integrated at the start to enhance sugar storage in sugarcane stem vacuolar organelle.

## 2.0. Materials and methods

### 2.1. Construction of basic and expression vectors with appropriate promoter system

The synthetic *ImSyGII* (1899bp) was cloned in plant expression vector, *pCAMBIA1301* to cause transgene expression in sugarcane (*Saccharum officinarum* L.). The BioBasic Pvt Ltd. provided services for chemical synthesis of codon-optimized *ImSyGII* and cloned in pUC57 vector. The *ImSyGII* (1899bp) was digested from pUC57 vector with enzymes BamHI and HindIII. The digested DNA fragment was ligated in *pCAMIA1301* vector using the same restriction sites. Three different ligation experiments were performed using single and double promoters. Firstly, the *pCAMBIA1301* expression vector (11,850bp) was cut with restriction enzymes *(KpnI-BamH1)* and single *Zea mays ubiquitin* (*Zm-pUbi*, 1977bp) was ligated. The *ImSyGII* fragment digested between *BamH1 and HindIII*,excised from the pUC57 vector, and ligated in *pCAMBIA1301 (Zm-pUbi-ImSyGII-NOS)*. In another cloning procedure, *Cestrum Yellow Leaf Curl Virus* (*pCmYLCV*) ligated plant expression vector was used for ligation of *ImSyGII* cassette *(pCmYLCV-ImSyGII-NOS)*. Thirdly, *pCAMBIA1301* vector with two constituted promoters *Zm-Ubi* and *pCmYLCV* were cloned with *ImSyGII* at *BamHI/HindIII* with *nopaline synthase* (NOS) as terminator *(Zm-Ubi-pCmYLCV-ImSyGII-NOS)*. The whole cloning procedure for constructing all three different types of the construct was carried out according to the methodological protocols [5, 20]. FastDigest restriction endonuclease enzymes were used for excision purposes, while DNA ligation and dephosphorylation of vectors were done by T4 DNA ligase enzymes [24] (Thermo Fisher Scientific Pvt. Ltd.).

### 2.2. Genetic transformation of sugarcane with different constructs

**2.2.1. Callus induction.** The sugarcane cultivar, HSF-240 was subjected to callus induction and used genetic transformation of *ImSyGII* in sugarcane. The leaf blades and sheaths were removed starting from the lower surfaces to the top of the dewlap leaves in order to achieve clarity for sugarcane explants. The sugarcane leaf rolls with 20–30 cm length, sterilized with 70% (v/v) ethanol were cut by sterilized scalpel blade and rinsed with water inside the laminar flow hood for consecutive 15 minutes. Immature leaf rolls were cut into pieces transversely into 1.0 mm thick sections after sterilization. The sterilized pieces of leaves and cuttings were submerged on Murashiage and Skooge induction Medium (MS medium with 3.0 mg/l of 2,4-dichlorophenoxyacetic acid [2,4-D]) for 30–35 days (for embryogenic calli) or MS0.6 medium (MS with 0.6 mg/l of 2,4-D) for 7–10 days (for embryogenic leaf roll discs). The calli plates containing MS media solidified with phytagel (6g/L) were ready for the particle bombardment experiment as described [25].

**2.2.2. Particle bombardment of single double and triple stacked promoter gene constructs by gene gun.** Fine tungsten particles (1.1 μm; Bio-Rad Laboratories, Inc.) were collected, weighed (1.0 mg) and sterilized with 70% ethanol (v/v) before the administration of DNA transformation protocol. The isolated recombinant DNA (5.0 μg) was quantified and

purified after growing from cultured recombinant bacterial strains. Different gene constructs containing single, double and stacked promoter systems adsorbed and coated on sterilized fine tungsten particle. The sodium salt with concentration, 1M NaCl and spermidine (14.0mM) were used for gene coating. All gene constructs were stained with 6x blue dye and poured down into the filter assembly of gene gun. Tungsten particles coated with *ImSyGII* constructs were shot with the support of an inflow gun, creating 24.0 inches Hg vacuum chamber and target distance was set at 8.0 cm. The first transformation event was accomplished after bombarding with pUbi/*ImSyGII*/Nos into HSF-240 calli. The second transformation event was done by bombarding *pCmYLCV* driven *ImSyGII* construct (*pCmYLCV*/*ImSyGII*/Nos). The third transformation event was performed with double promoter construct (*pUbi+pCmYLCV*/*ImSyGII*/Nos) which was bombarded with the particle gene gun. In the last experiment, all three constructs were transformed in one event stacking. The four transformation events were designated as STE1, STE2, DPTE and TCTE respectively as drawn in Fig 1.

## 2.3. Bombarded calli regeneration and screening of putative transgenic sugarcane plants by drug selection

Immediately after bombardment, MS calli plates were kept at light under ±28°C for 11 consecutive days so that calli plates started regeneration as in previous methods [11]. After 11 days, regenerated calli were shifted to selection media (MS3 and MS4 with hygromycin (50mg/ml) selection for 15 consecutive days in dark conditions while maintaining the same temperature [18]. Shoot regeneration media (MS salt, 4.12 g/L, Sucrose, 20 gms/L, Myoinositol, 0.50gms/L supplemented with Gibberellic acid3, 2mg/L 2.5mg/L kinetin after autoclave with PH

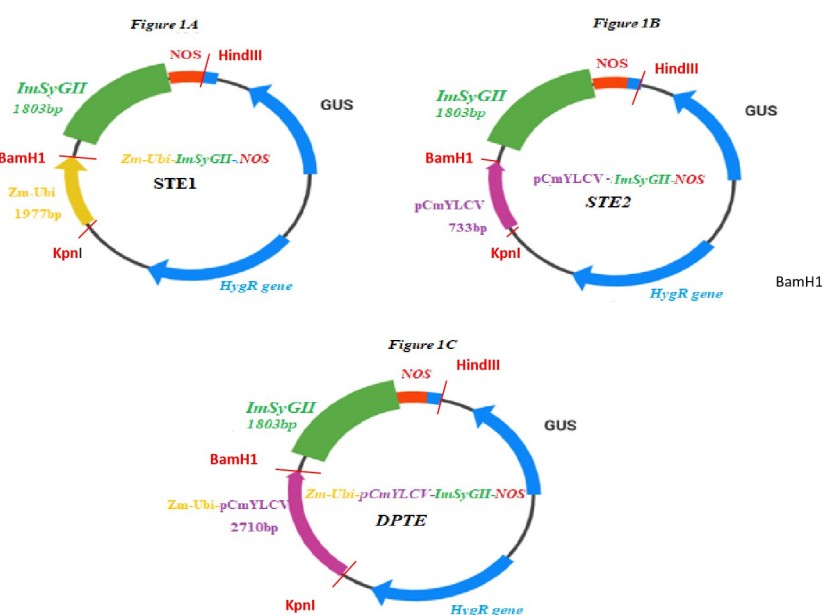

**Fig 1.** (A) Illustrations of the genetic map with single transformation event 1 (STE1), driven under ubiquitin expressing *ImSyGII* in the expression vector (B) Genetic vector map indicating the cloning of *ImSyGII* under the influence of *pCmYLCV* terminated by NOS. (C) Plant expression vector map exhibiting inclusion and integration of *ImSyGII* (1899bp) driven under dual promoter regions *Zm-Ubi-pCmYLCV* and terminated by NOS. Both promoters, Zm-Ubi and pCmYLCV ligated in upstream region of the gene construct *ImSyGII*. Their presence on upstream region helps in enhancing the expression of the *ImSyGII*. The ligation of promoters was done between KpnI and BamHI while DNA fragment, *ImSyGII* was ligated between BamHI and HindIII restriction sites. All three genetic constructs employed hygromycin (100mg/mL) as a selection marker.

maintained at 5.7) and root generation media (MS salt, 4.12g/L, Sucrose, 20g/L, Myoinositol, 0.5g/L supplemented with Naphthalene Acetic Acid (NAA), 2.0mg/L, Indole-3-Acetic Acid (NAA), 3mg/L at pH 5.7) along with hygromycin (50mg/ml) marker were prepared. The transformation efficiency of regenerated transgenic sugarcane plants was also determined at this stage. The total number of putative transgenic plants were divided by total number of regenerated regenerated plantlets were calculated. The number of survived putative transgenic plants under selection media was divided by total number of regenerated putative transgenic plants and multiplied by 100. After two weeks, plantlets survived on selection media were transferred into potted soil in controlled environment. The hygromycin selection of callus and regenerated plantlets was executed to screen putative transgenic and non-transgenic plants. The control plants were not treated with selection media and grown naturally.

## 2.4. DNA extraction and PCR amplifications based screening of transgenic plants

Fresh leaf samples from putative transgenic leaves were collected wrapped up and labeled properly dipped into liquid nitrogen bucket, .Frosted leaves were immersed in liquid nitrogen and crushed into fine powder using pestle and mortar. The collected samples were put into 1.5 ml eppendorf tube. Preheated extraction buffer solution containing 20mM EDTA, 100mM Tris-HCL, 2% CTAB, NaCl 2M, 1% Mercaptoethanol with 0.2g PVP/g leaf powder were poured into 1.5ml eppendorf tubes containing powdered leaf samples. The suspension powdered sample was incubated at room temperature 65°C for 60 minutes and shaking was done intermittently. After one hour, samples were extracted out of the water bath and cooled down to room temperature. The mixed solution of chloroform: Isoamylalcohol having ratio 24:1 with 10ml volume was added and inversion was done gently. Samples were centrifuged at 14000g at 4°C for 5 consecutive minutes. The supernatant was collected and poured in new and fresh Eppendorf tubes. The repetition of the previous step was executed time and again until the aqueous phase became clear. The addition of 5M, 1/5th volume of NaCl with complete mixing and incubation with 2-propanol at the temperature of -20°C was done overnight resultantly maximum DNA precipitation was accomplished. Next day, immediately the centrifugation at 12000 g for 10 min was accomplished at room to get DNA pellets. The resultant pellets were washed initially with 70% ethanol, air dried and DNA pellets were dissolved with 1/10th of TE buffer. The RNA contamination from DNA was removed by adding RNAs (10μg/ml) and finally put at 37°C incubator for 30 minutes. The Phenol was mixed in equal volumes into DNA, centrifugation was accomplished at 3000 g for 5 minutes. The extracted DNA was evaluated qualitatively by 260/280nm absorption ratio and checked by agarose gel electrophoresis. A polymerase chain reaction (PCR) experiment was executed to confirm the presence of transgene *ImSyGII* in transgenic sugarcane plants with single, double, and triple constructs. Primer sequences as follows, *ImSyGII-F*: CAGTCGCGACATACAGCAAA and *ImSyGII-R*: GGCACCCC GAATATTTCACC. The DNA template was amplified in total reaction volume 20μl by exploiting 100ng DNA using Taq DNA polymerase according to manufacturer's instructions. Following instructions and reaction conditions were set at C1000 Touch thermal cycler (BioRad Labs.): 95°C for 4 min, 35 cycles each at 94°C for 30s, 52–58°C for 45s, and 72°C for 7 min. Full-length primers were designed from Primer 3.0 online software tool covering the whole sequence range from promoters, *ImSyGII* and terminators. Gel electrophoresis was performed to separate amplicons at 2.5% agarose gel (w/v) stained with ethidium bromide. Negative control (no DNA template) was also run for PCR.

## 2.5. Estimation of *ImSyGII* expression in transgenic sugarcane by real-time PCR

Fresh leaf and stem samples from PCR positive transgenic plants were taken to extract RNA. The RNA extraction was done by using TRIZOL method. Fresh samples were crushed into fine powder in liquid nitrogen and powder was collected in new 1.5mL eppendorf tubes. Trizol (1mL) was added in it and mixed thoroughly. The homogeneity samples was separated into liquid and organic phases after the addition of chloroform. The centrifugation was done to keep RNA in the upper phase. The extracted RNA is allowed to precipitate by adding isopropanol which was followed by centrifugation to make pellets of the RNA. The RNA pellet is washed with 75% ethanol to remove impurities, air-dried, and dissolved in RNase-free water or an appropriate buffer for downstream applications. The DNAse was also added to denature any DNA present in the samples. This method efficiently isolates high-quality RNA suitable for various molecular biology techniques. The extracted RNA was converted to cDNA by using thermo-scientific kit (cat. #). The Real-time PCR (qPCR) was administered to determine expression of genes with single and dual promoters with *ImSyGII* in a transgenic sugarcane line. The primer used in this work was as 5'AGGCCGTCTTCTATCAGGTG3'. The model CFX384 Real-time PCR detection system (BioRad Laboratories) with SYBR Green mixer was used. Each specific target primer with 0.4 μM and 1.0ng of genomic DNA from candidate (*ImSyGII*) transgenic sugarcane was used. The Real Time PCR experiment with proper conditions were set which were as follows: denaturation at 95.0°C for 3 min, 39 two-step cycles each at 96.0°C for 5s and 57°C for 30s, and a final melting curve of 60.0°C to 95.0°C for 6 min. The *Beta-actin (β-actin)* gene was used as an internal or reference gene. The PCR amplification efficiency was calculated with LinReg [19] Threshold cycle ($C_T$) values. The Relative Expression Statistical Tool (REST) was used to explore expression level through recorded $C_T$ values in different samples. The control sample with non-significant value was set as standard.

## 2.6. Estimation of sugar recovery percentage from BRIX method

Screened PCR-positive transgenic sugarcane lines were subjected to sugar estimation at 12 months and 18 months of maturity. Transverse stem samples were taken from the top, middle and lower portions so that the sample might represent sugar samples from the whole internode. Sugarcane stems were chopped to sizes of 30cm by crushing of sugarcane samples by shatter machine. The screening of small sugarcane stem sizes between 2-4mm was obtained. The crushed extracted cane juice samples were subjected to FT-NIR (MPA, Bruker Optics, Ettinger, Germany) spectrometer analysis. The five hundred grams of shattered sugarcane were rotated in a drum with 250mm diameter. The diffuse reflectance mode was set at 27°C for good analysis of sugar estimation. The refractometer (RX 5000 CX, ATAGO, Japan) was employed to estimate Brix with 0–60% at the temperature of 22°C. The measurement of juice polarity % age in cane samples was done after mixing 50mL cane juice along with 8 mL Aluminium sulfate solution (200g/L) and 5mL sodium hydroxide (2M). The final volume of the sample was made up to 200mL. The resultant solution was mixed well and filtration was completed through filter paper (45mm in diameter, 11μm pore size thickness, 180μm). The filtrate was already employed for the determination of optical rotation through polarity. The calculated Brix and OR also help determination of polarity %age by the following equation:

$$Pol\,(\%) = \frac{-6.51 + (25.3 \times OR) - 0.011\,(OR)2 + \left(2.93 \times {}^0Brix\right) - 0.207\,\left({}^0Brix\right)2}{100}$$

The above equation represents Optical rotation (OR) while Brix is the concentration of soluble solids in cane juice. Moreover, the fiber content of each sugarcane line was determined by

the following formula:

$$\text{Fiber \% (FC)} = \frac{(1002 - 100 \times \text{moisture \%} - 97 \times \text{Brix})}{(100 + \text{Brix})}$$

The commercial cane value (Sugar recovery %age) was evaluated from all above-given values of Pol (%), Brix (%) and Fiber (%) by the following equation:

$$\text{CCS\%} = \frac{0.943(pol\%)(100 - FC\%)}{100} - \frac{1}{2}0.943\frac{(Brix\%)(100 - FC\%)}{100 - (pol\%)(100 - FC\%)\frac{100-FC\%}{100}}$$

## 2.7. Estimation of IM and sugar contents from transgenic sugarcane stems by HPLC

Transgenic sugarcane lines which exhibited maximum sugar levels obtained after Brix, the further step was to quantify sugar contents through HPLC. The fresh cane juice from all sugarcane lines was extracted from each internode to show true representation of the whole internode. The dilution of cane juice was executed by 1:2 employing acidified deionized water (pH 2.0). All required chemicals including acetonitrile, a gradient HPLC, ammonium formate, ammonium bicarbonate, ammonium acetate, formic acid, acetic acid, standards of glucose (GLU), FRU fructose, IM and TH were purchased from Sigma-Aldrich chemicals. (Merck Germany). The instrumentation comprising HPLC Dionex Ultimate 3000 system having variable LC 380, consists of evaporative light scattered through detector. The HPLC column (HALO Penta-HILIC, DE USA) with dimensions, 150×4.1mm, 2.8μm which was also used as a stationary phase. The ammonium formate filled with acetonitrile (40mM) also used as mobile phase. The elution consisted of acetonitrile, and ammonium formate with (40mM, pH, 3.7) respectively. The mobile phase specifications which was set at the flow rate of 3.0 mL/min with temperature (10˚C) and 4.0μL injection volume. The final detector, ELSD with the following specifications: width 10mm, gained 2. The evaporation was set at a temperature of 40˚C, with normalising temperature 40˚C and nitrogen flow rate 1slm. The MS Excel was employed to execute statistical calculations including standard and relative deviations.

## 3.0. Results and discussion

### 3.1. Confirmation of *ImSyGII* under single, dual, and triple promoters driven transgenic sugarcane lines by PCR

Three different cloning experiments were carried out in the present study as already described in the materials and methods section. The integration of *Zm-Ubi* and *pCmYLCV* separately and in combination were embedded in the plant expression vector. It created differential promoter-based constructs further confirmed by restriction digestion experiment. The *Zm-Ubi* promoter was integrated between *KpnI* and *BamHI* restriction sites and yielded 1993bp fragment upon digestion as illustrated in Fig 2A. Fig 2B illustrating gel pictures with resultant bands, showing successful cloning of *ImSyGII* under *Zm-Ubi* and *pCmYLCV*. Third cloning experiment in which two different promoters, *Zm-Ubi* & *pCmYLCV*, drove the expression of *ImSyGII* and embedded successfully. Fig 2C represented this combination of promoters by restriction digestion and producing a 2710bp band after successful cloning. The Fig 2D illustrated the PCR amplifications of *ImSyGII* gene in all combinations of promoters.

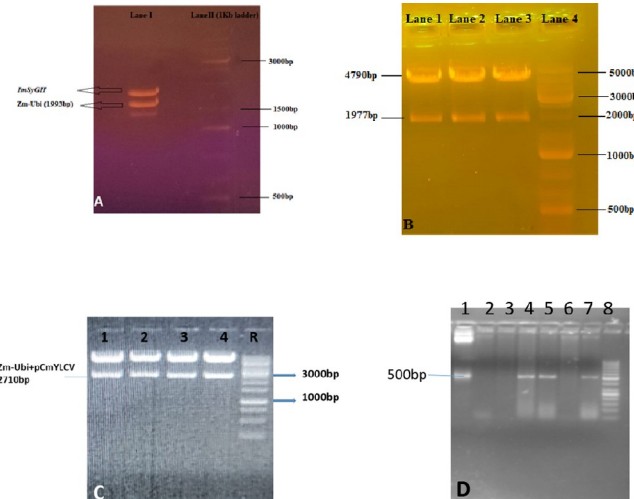

**Fig 2. Illustrations of various strategies designed for cloning with single, double and multiple promoters driven ImSyGII.** (A). Restriction digestion results confirmed the cloning of *ImSyGII* in a vector controlled by the promoter region of *Zea mays* ubiquitin. Two fragments in the gel at their desired locations are evidence of successful cloning. (B). Restriction digestion results denoting dual promoters (*Zm-ubi-pCmYLCV*) along with the gene of interest *ImSyGII* fragmentation cut from original vector. (C). Restriction digestion results show clone in lane 1 while lane 2 indicates a 100 bp gene regulator. (D). PCR amplification (500bp) amplicons of *ImSyGII* in three different constructs shown by lanes (1–7) containing positive control (1), negative control (2) and with positive samples (4,5,7) run against a 1kb ladder.

## 3.2. The promoter *Zm-Ubi* triggers *ImSyGII* more efficiently in leaves than stems, while *pCmYLCV* appears highly operational in stem tissues

Quantitative RT-PCR analysis of transgenic sugarcane lines was subjected to expression level estimation in various lines. The *ImSyGII* transformed lines expressed differently in leaves and stem tissues. Single transformation event lines (STE1) exhibited differential transcriptional levels in sugarcane lines as shown in Figs 3 and 4. The selected lines were shown in graphical representations with variations in different tissues. Transgene *ImSyGII* is expressed more robustly in great quantity as compared to stem tissues. STE1-8, STE1-20, STE1-26 and STE1-22 lines exhibited 2.7, 2.7, 2.6, and 2.6 times higher expression levels than control lines, respectively, reflecting 66.6% and 57.7% boosted expression than in stem tissues. Similarly, transgenic lines STE1-14, STE1-17, STE-1-31, STE-1-33, STE1-32 and STE-1-1 denote 2.5, 2.5, 2.2, 2.1, and 2.3 times increased *ImSyGII* expression levels respectively in leaf tissues. These lines also reflected 60.8%, 60.86%, and 66.6% enhancement than expression shown in stem tissues when sugarcane transgenic lines reach 12 months of age. After six months, fully matured sugarcane lines STE1-1, STE1-33, and STE-1-22 manifested 84.2%, 83.3%, 105% and 94.7% increase in transgene expression in leaf tissues than in stem tissues. While STE1-8, STE1-14, STE1-17, STE1-20, STE1-31, and STE1-32 transgenic lines showed enhancements of 65.2%, 71.4%, 59%, 60.8%, 75% and 63% in leaf tissues than in stems respectively. Reported outcomes favour the inclination of *Zm-Ubi* promoter in leaf tissues than in transgenic sugarcane stems. The following Fig 5 illustrates the above-described results efficiently. The statistical data with requisite information is provided in supplementary file S-5A,S-5B,S-5C and S-5D.

## 3.3. The promoter *pCmYLCV* appears highly efficient in producing increased expression in transgenic sugarcane stem tissues

In the second sugarcane transformation event (STE2), *pCmYLCV* was exploited to trigger the expression of *ImSyGII* measured by quantitative RT-PCR analysis in both leaf and stem tissues.

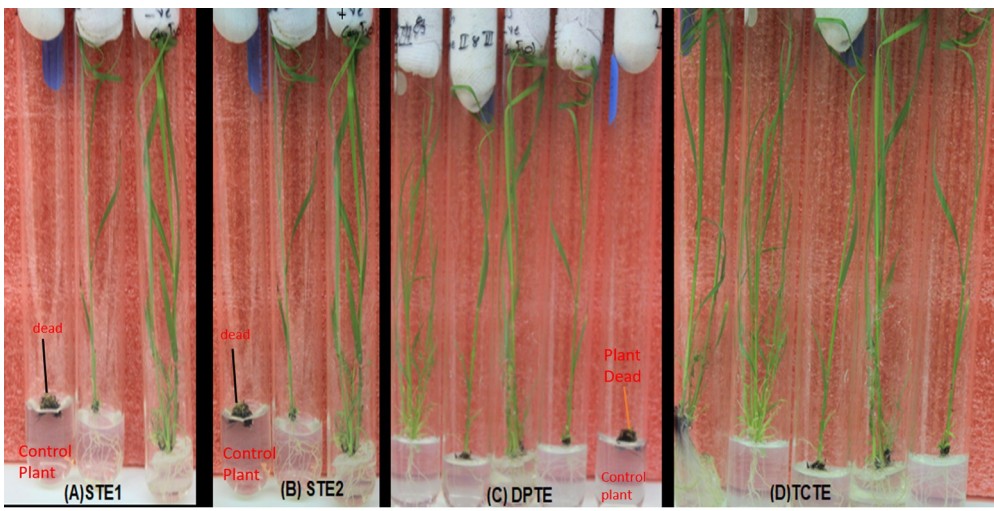

**Fig 3. Screening of putative transgenic sugarcane plants under hygromycin selection.** (A). It demonstrates putative transgenic sugarcane plants from STE1 show good growth and healthy phenotype while non-transgenic plants were dead in hygromycin selection. (B). STE2 plants from *pCmYLCV* promoter driven *ImSyGII* expression shows presence in some plant while non-transgenic sugarcane plants become dead. (C). Dual promoters drove *ImSyGII* putative positive sugarcane plants to survive under hygromycin while non-transgenic plants could not survive. (D). Putative positive TCTE plants screened under hygromycin selection.

Data revealed a different pictures as obtained in STE1 after 12 and 18 months. Transgenic lines STE2-35, STE2-30 and STE2-31 after 12 months exhibited *ImSyGII* transcriptional activity of 4.8, 4.6 and 4.3 times increased than internal line STE2-1. These lines appeared more active with 100%, 109% and 114% mounted expression levels, respectively, in stem tissues compared to leaf tissues. Some expression data inferred from Fig 6 manifested STE2-3, STE2-39, STE2-36, STE2-6 and STE2-8 produced 3.9, 3.8, 3.7, 3.7, 3.7 and 3.6 times higher than internal control lines. Similar transgenic lines exercised 105%, 137.5%, 105%, 131.2% and 140% higher *ImSyGII* activity in stems than leaf tissues. The remaining lines driven by

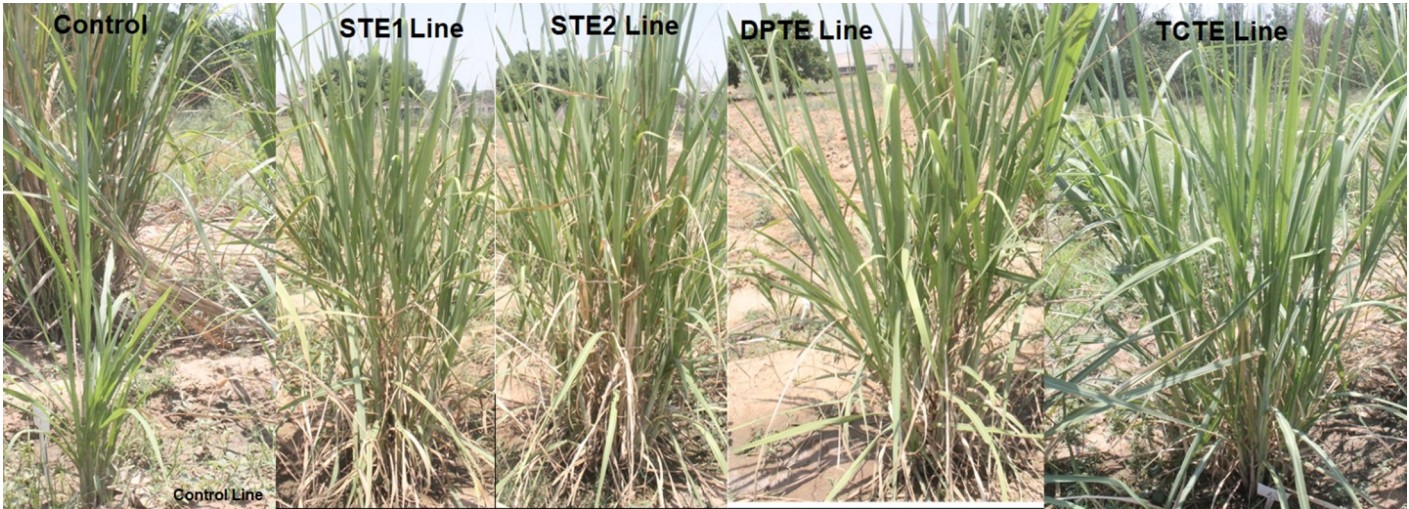

**Fig 4. PCR-positive transgenic sugarcane plants from STE1, STE2, DPTE and TCTE, along with non-transgenic sugarcane plants grown after 6 months under field conditions, are shown in this diagram.**

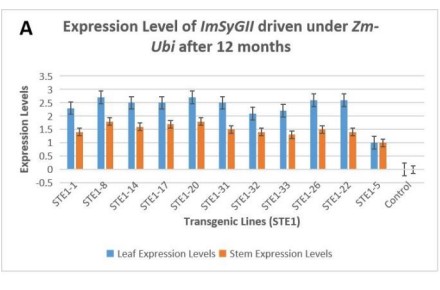

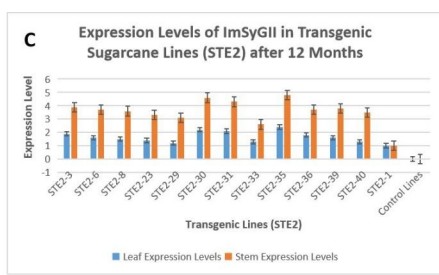

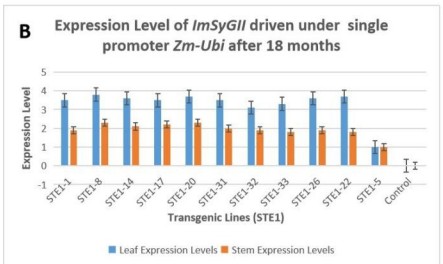

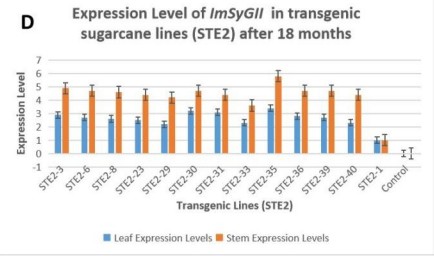

**Fig 5.** (A). Manifestations of different transgenic lines with STE1 construct illustrating differential expression profile for *ImSyGII* in leaf and stems after 12 months (B). Illustrations of STE2 harbored transgenic sugarcane lines manifesting expression levels of *ImSyGII* after 12 and 18 months old plants in leaf and stem tissues. (C). Expression levels of *ImSyGII* under the influence of ubiquitin promoter in leaf and stems after 18 months of age. (D). Graphical representations of *ImSyGII* expression levels in leaf and stem tissues harboring under constitutive promoter *pCmYLCV* after 18 months. The statistics of the raw data in all categories are available in supplementary files S-7A-D.

*pCmYLCV* promoter STE2-23, STE2-29 and STE2-40 were obsessed with 3.3, 3.1, and 3.5 times enhanced *ImSyGII* gene expression in stems than leaves, respectively. After 18 months, mature sugarcane lines STE2-35 manifested the highest level of transgene expression, inferred as 5.8 times higher than internal lines in sugarcane stems, while STE2-33 stands at 3.6 times higher. Fig 6 illustrates and summarizes the performance levels of all transgenic lines with the STE2 construct. The expression level was significantly enhanced in stems than in leaf tissues. Lines STE2-35, STE2-3, STE2-6, STE2-8, STE2-23, STE2-29, STE2-30, STE2-31, STE2-33, STE2-36, STE2-39 and STE2-40 documented percentage increase enhancements in expression levels stood at 70.5%, 68.9%, 74%, 76.9%, 91%, 46.8%, 41.9%, 56%, 70.5%, 67.8%, 74% and 87.3% respectively.

## 3.4. The DPTE yields increased *ImSyGII* expression levels than STE1 and STE2

The *Zm-Ubi* promoter in STE1 events shows different expression patterns of *ImSyGII* in multiple positive transgenic lines. This research work manifested the combinatorial promoter system and its regulation in enhancing *ImSyGII* expression which was highly commendable. Similarly, the DPTE lines showed increased *ImSyGII* expression after 12 and 18 months of age. The DPTE-37 exhibited 5.5 times level of higher *ImSyGII* expression in sugarcane stems as compared to internal control line while DPTE-23 showed 4.6 times higher expression than internal line. Similarly, DPTE-5, DPTE6, DPTE-7, DPTE-9, DPTE-15 and DPTE-16 reflected 4.9, 5.1, 4.9, 5.2, 5.1 and 4.9 times higher *ImSyGII expression* in control lines respectively. These sugarcane events also showed 36%, 34%, 32.4%, 36.8%, 30.7% and 29% increased expression respectively than in leaf tissues. The DPTE-35, DPTE-38 and DPTE-39 were intrinsically 5.2, 5.2 and 5.3 times strongly expressed *ImSyGII* with 38.4%, 26.8% and 23.2%

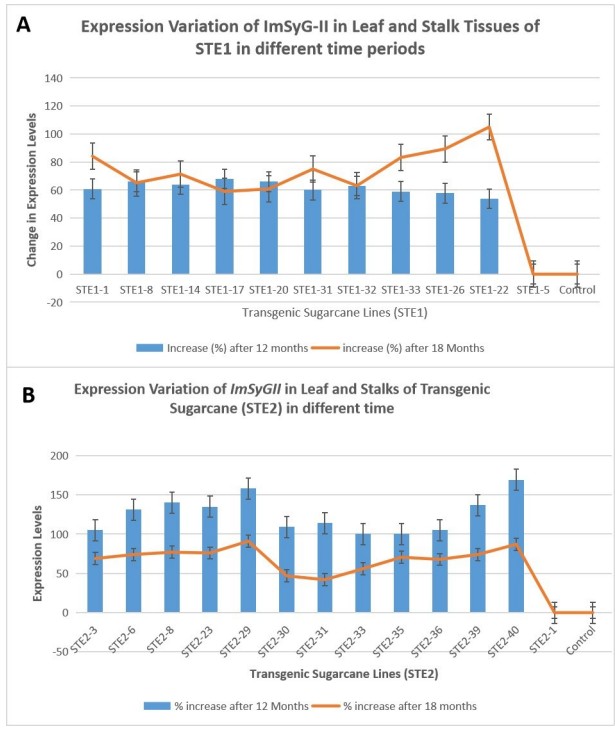

**Fig 6.** (A). A comparison was drawn among lines regarding their change in expression levels about STE1 in leaf and stem tissues with different time periods. The orange line denotes the percentage enhancement of transcription after 18 months, while the blue bars represent an increase after 12 months. (B). Graphical representation of the STE2 expression comparison among various transgenic lines in their leaves and stems in intervals of 6 months. The orange line shows an increase in expression level after 18 months, while the blue bars highlight the expression profile after 12 months. The statistics of the raw is available in supplementary file as S-7A and S-7D.

respectively which was higher than leaves after 18 months as shown in Fig 7. The statistics of the whole raw data has also been provided in supplementary file as S-7A, S-7B, S-7C and S-7D.

## 3.5. Highest levels of *ImSyGII* in triple stacked promoters constructs transformation event lines

Triple construct transgenic event lines (TCTE) manifested all-time higher expression than control ones. Transgenic lines subjected to RT-PCR after 12 months of age confirmed the highest expression line was TCTE-13, with 5.8 times more expression than TCTE-1. Twelve months old transgenic lines indicated more than 30% increase in expression in stem tissues than in leaf tissues. TCTE-8, TCTE-12, TCTE-13, TCTE-15, TCTE-19, TCTE-25, TCTE-31, TCTE-35, TCTE-38 and TCTE-40 triggers 28.5%, 32.5%, 28.8%,30.7%, 32.5%, 34.1%,28.5%, 31.5%, 35.9% and 35.9% enhancements in *ImSyGII* expression level in stems than in transgenic leaves. After an 18 months similar percentage increase in the expression profile of sugarcane stems was observed than in leaves. The highest percentage enhancement level was observed in the TCTE-40 line, which was 50% more in stems than in leaves. In comparison, TCTE-13 showed 6.7 times higher expression in stems after 18 months than internal control line TCTE-1 as shown in Fig 8.

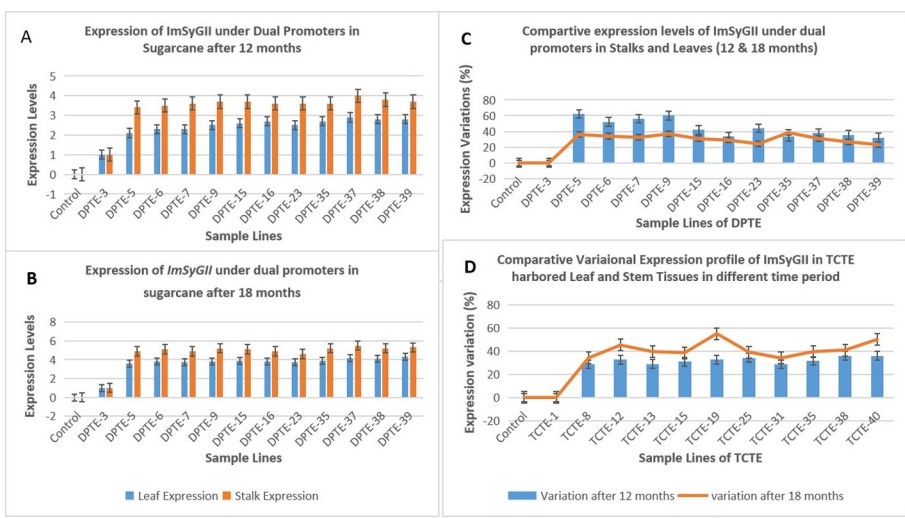

**Fig 7. Illustrations of expression levels of DPTE after 12 and 18 months old sugarcane plants in their leaf and stem tissues.** (A). This figure can describe DPTE expression among major transgenic lines after 12 months in leaves and stems (B). Expressions of DPTE construct in various transgenic lines after 18 months of age in sugarcane. (C). The comparison is drawn among transgenic lines in their leaves and stems in response to their degree of expressing DPTE after 12 and 18 months of old plants. (D). Variations of *ImSyGII* expression in TCTE construct among major transgenic lines after 12 and 18 months.

## 3.6. Expression levels increased significantly in mature internodes than in juvenile sugarcane lines

The STE1, STE2, DPTE and TCTE sugarcane lines manifested increased levels of ImSyGII expression in mature internodes. The *ImSyGII* expression in 12 months old transgenic

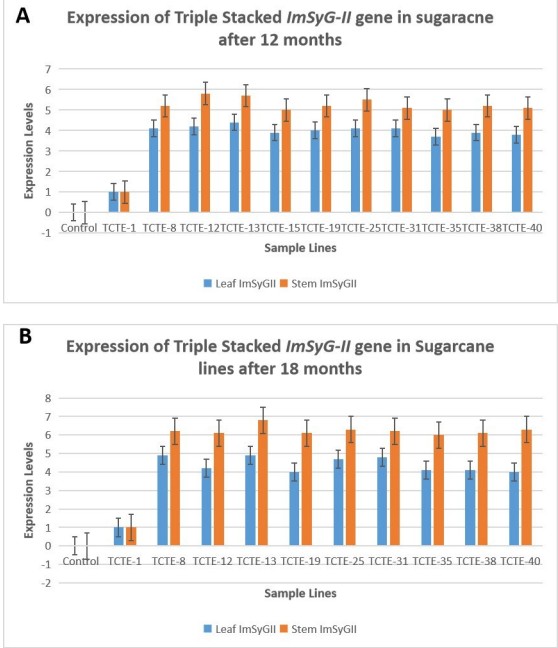

**Fig 8.** (A). Illustrations of expression levels shown by three stacked gene constructs transformed into sugarcane plant individually after 12 months both in leaf and stem tissues (B) Expression levels of triple stacked gene constructs showing *ImSyGII* gene in both leaf and stem tissues after 18 months transgenic sugarcane lines. The supplementary file contains statistical information of the raw data as S-8A and S-8B.

sugarcane lines witnessed lower *ImSyGII* expression levels as compared to mature internodes (18 months). Significant enhancements of *ImSyGII* expression driven under different constitutive promoters observed in younger and mature sugarcane internodes and in leaves. The Figs 6–8 illustrated the findings very efficiently. The STE1-8 exhibited 2.7 times higher expression level in 12 month old plant further increased to 3.8 times in leaves. Likewise, STE1-20 showed 3.7 times in leaves which were observed to be 1.7 times at 12 months. In STE2 lines, transgenic sugarcane lines STE2-23 had 3.1 times at 12 months which increased to 4.2 times at 18 months. DPTE-37 line also manifested a significant increase in mature internodes (5.5 times) relative to 4 times in 12 months. Similarly, the DPTE-39 line expressed 5.3 times higher in mature sugarcane lines compared to 3.7 times in juvenile sugarcane lines. Triple stacked promoter constructs with TCTE-15, TCTE-35, and TCTE-19 showed expression levels 5.1, 5 and 5.3 times, respectively, which triggered to be enhanced to 6.1, 6 and 6.2 times, respectively, after 18 months of maturity. The synchronized enhanced *ImSyGII* expression in mature sugarcane lines was obvious in all transgenic sugarcane lines, as shown in Figs 7 and 8.

## 3.7. Expression of *ImSyGII* under different constitutive promoters depends upon target tissues

It was also observed from experimental data obtained from the present study that different tissues behave differently in transgene expression. Transgenic lines harboring STE1 construct driven under the *Zm-Ubi* promoter manifested higher expression in leaves than in stem tissues (Fig 6). Similarly, the STE2 construct having *pCmYLCV* promoter was highly effective in sugarcane stems than in leaves (Fig 7). Results shown in various figures indicated that some constitutive promoters (*Zm-Ubi*) were more effective in leaves, while other double promoters (DPTE) and triple stacked promoters (TCTE) revealed boosted transgene expression levels in sugarcane stems than in leaves.

## 3.8. Highly expressed transgenic lines manifested increased sugar levels upon sugar analysis

PCR confirmatory transgenic sugarcane lines were subjected to estimation of sugar contents through the BRIX method. Sugarcane lines from control, STE1, STE2, DPTE and TCTE were employed to estimate polarity (%), Fiber, Brix and commercial cane sugar (%). Results showed that less sugar recovery and Brix percentages obtained from control samples reported to be 10.5% on average. Fiber percentage remains constant for all sugarcane lines (control and transgenic). Sample lines from all groups driven under *Zm-Ubi*, *pCmYLCV*, dual promoters and triple-stacked promoter constructs harboring sugarcane events were taken for estimation. Fig 9 indicates that TCTE lines harvested the highest Brix and CCS % among reported lines. The DPTE lines performed much better than STE2 & STE1 lines. While the STE2 lines with little improvements showed better outcomes than STE1 lines. All STE1 lines outpour better sugar contents than control lines as showed in Fig 9.

## 3.9. HPLC explored SuperSugar lines in TCTE category, producing augmented levels of IM without diminishing sucrose levels

Sugarcane transgenic lines, which testified and produced higher sugar recovery percentages than control lines according to data obtained from Brix estimation, were subjected to HPLC analysis. Target was to explore internode level estimation of all major sugar contents. (GLU), fructose (FRU), sucrose (SUC), Isomaltulose (IM) and trehalulose (TH) were extracted and quantified as per criteria defined in HPLC protocols. Transgenic sugarcane lines STE1-8,

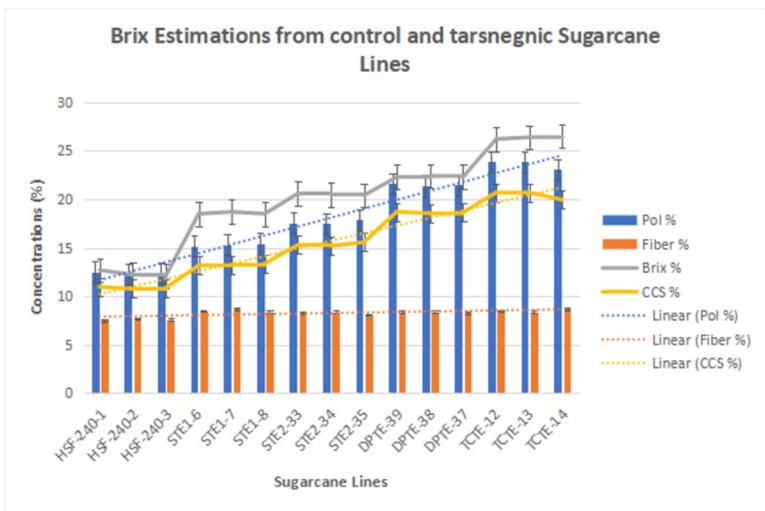

**Fig 9. Graphical representations of the concentrations (%age) of different sugar components extracted from various transgenic and control lines (HSF-240) via the Brix method.** Blue bars showed polarity percentage values, orange bars indicate fiber %age present in sugarcane lines, Grey lines present a trend of Brix percentage values, while yellow lines show a percentage increase of commercial cane sugar concentrations in control as well as transgenic internodes. The statistical information of the provided raw data is available as S-9 in supplementary file.

STE2-35, DPTE-39 and TCTE-13 were selected, and their whole-length internodes were evaluated for sugar contents estimation. All graphical figures testified the highest level of IM yield as shown by yellow lines. Other sugar contents, GLU, FRU and SUC remain intact, and their concentrations remain more or less identical in all different transgenic events. While IM contents gradually increased significantly throughout the entire length of the transgenic sugarcane. There was a significant enhancement in IM concentrations in the stacked TCTE-13 line, reaching a maximum value of 648mM. While STE1-8, STE2-35 and DPTE-39 showed IM values at 301mM, 467mM and 564mM, respectively as shown in Fig 10.

## 4.0. Discussion

The complexity of genome pose serious challenge in transformation of foreign gene as well as its integration in sugarcane [26]. The present work not only evaluated the performance of more than two promoters and their combination but also succeeded in improving overall sugar contents in existing sugarcane varieties. Furthermore, the *ImSyGII* synthesized sucrose isomer, IM in transgenic sugarcane line which produced highly sweet and healthful sugars in cane juice. The genes which encode sucrose isomerases (SI) were retrieved from different micro-organisms especially bacteria, whose sequences were present in NCBI. The cloning of *ImSyGII* was done in plant expression vector under different promoter combinations and transformation was implemented by particle gene gun method [27]. In previous work, transgenic line UQ68J was transformed with SIG from *Pentoea dispersa* which doubled the sugar contents as compared to control lines. The genotyping shows that signal peptides NTPP were also integrated in upstream region of promoter which help directing proteins into vacuole [22]. The work done by Wu and Birch, 2007 witnessed substantial IM variations among transgenic lines as shown by HPLC. The physiological studies also revealed that the sucrose contents doubled in next vegetative generations while IM levels declined in successive generations. Such decrease in IM levels in sugarcane happened primarily due to invertase activity in sugarcane cells [28]. The present work transformed with *ImSyGII* codon optimized

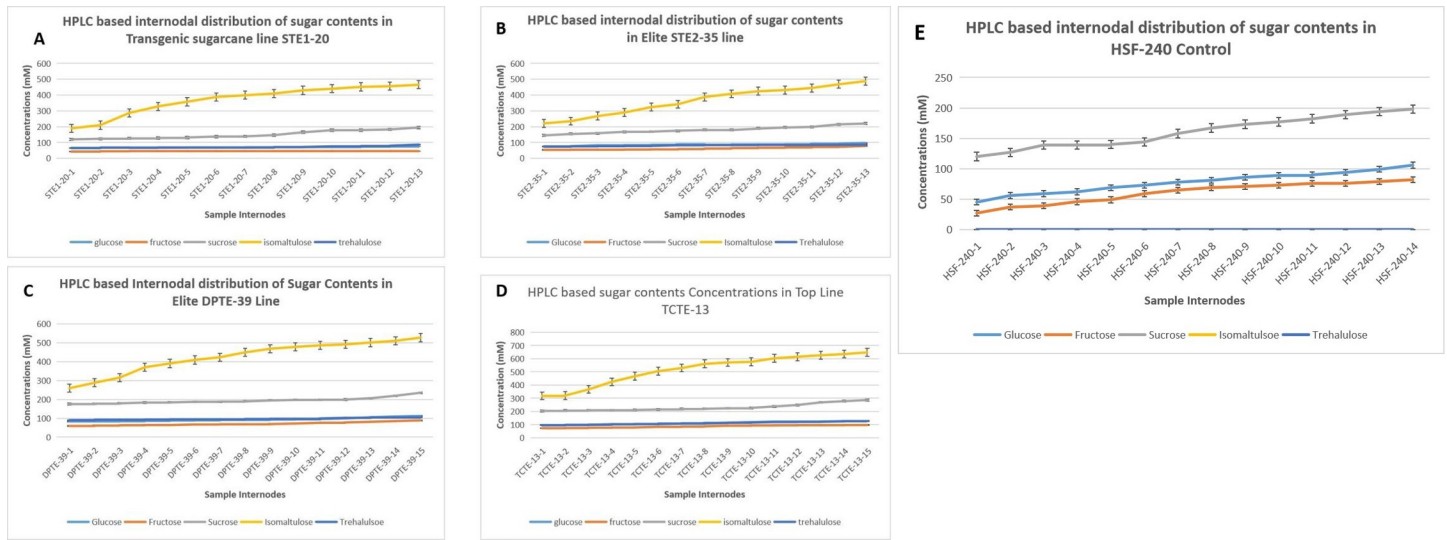

**Fig 10.** (A). Distribution of sugar contents in internodes of top STE1-8 transgenic sugarcane line, quantified from High-Performance Liquid Chromatography (HPLC). (B). HPLC-based sugar contents distribution in internodes of elite sugarcane line STE2-35. (C). Illustration of best sugar line DPTE-39 regarding sugar contents distribution in its internodes after HPLC. (D). Estimating various sugar contents and their quantifications via HPLC in internodes of Supersugar line (TCTE-13). The light blue line shows GLU, the orange line denotes fructose concentration, the yellow line represents IM the grey line indicates sucrose molecule, and the dark blue line represents trehalulose. (E). The distribution of sugar contents in control line HSF-240 were also drawn in this figure. The absence of yellow and blue lines in this graph showed that no IM and TH were obtained from control lines after HPLC analysis. The statistical information of the raw data is available as S-10A, S-10B, S-10C, S-10D and S-10E.

from *Erwinia rhapontici* reflected better harmony among transgenic lines in terms of IM concentration as quantified by HPLC. The sucrose contents very much likely to be unchanged in transgenic lines while the IM concentration keep on enhancing from one internode to lower ones. The highly increased sugar contents from control lines was happened to be due to the presence of vacuole targeted signal peptide [29]. The *ImSyGII* positive transgenic sugarcane lines with integrated vacuole targeted sequence (VTS) driven under *Zm-Ubi* and *pCmYLCV* expressed much similar phenotypes in terms of morphological features even somewhat close to control lines. The cane juice with STE1 showed less sugar concentrations in young and mature internodes as compared to STE2, DPTE and TCTE lines. Previous studies adopted cloning of SI under only ubiquitin promoter and yielded IM in the range of 39–53%. The present study produced TCTE-13 lines, showing with IM quantifications stood 45–61% of the total sugar contents while, SRP levels in transgenic lines increased to 75% than control lines. Similarly, DPTE-39 indicated 41–57.7% IM from the total sugar concentration of the cane juice. The sucrose values remain unchanged in transgenic as well as control lines, owing to the presence of vacuolar signal peptide in the constructs. Such a huge variation and enhancement of IM concentration lines might emerge due to the difference in gene copy numbers among transgenic independent lines. Another past work was focused on the development of TH producing sugarcane transgenic lines which also ultimately enhanced sugar contents but the concentration of IM was less than TH while the present work witnessed TH ratio less than 1% of total sugar values which was more than 90% of sugars in TH transgenic lines previously. Although, both sucrose isomers depict the increased healthful sugar values but may represent the genetic differences, arises due to evolutionary difference among source microbial species [11]. Less Sugar recovery percentage from local sugarcane crop varieties is the biggest issue worldwide. The *ImSyGII* was introduced under various single, double and triple promoter systems to ensure increased sugar recovery percentage in sugarcane. Data from Figs 8–10, 12 and 13 seemed a very promising effort to enhance overall sugar contents in sugarcane crops. Some

vacuole-targeted signal peptide proved helpful conductors of sugar storage parameters in the compartmentalization of sugar in stems vacuoles. Interestingly, similar research was performed to transform sucrose isomerase genes in *Saccharum officinarum*, and HPLC-based data determined sugar contents. Such studies revealed the doubling of sugar contents in contrast to our studies which showed an 80% enhancement in sugar recovery percentage [30]. Another research was conducted to evaluate seven recipient sugarcane genotypes for over 3 years in field conditions in Australia. Such studies witnessed the capacity of IM stood at 33% of total sugars obtained from cane juice, while present studies revealed 53% IM in overall sugar contents. That studies witnessed an overall decrease in sugar contents while total sugar contents remained intact [34]. The present study employed more than two promoters hence, increased transgene expression leads to enhanced sugar recovery percentage without decreasing sucrose contents [31]. Sucrose is not affected by *ImSyGII* because acidic conditions in vacuole protect breakdown of sucrose which remains stored inside vacuole, owing to the presence of vacuole-targeted sequence signaling [28]. The working of *ImSyGII* requires alkaline environment which is easily provided by cytosolic region but in vacuolar region due to acidic environment, the enzyme Isomaltulose synthase cannot work at optimum rate and hence, the substrate sucrose remains intact. This extra SUC storage in vacuole also added up to final IM and collectively enhanced the final sugar production [32]. The fate of IM is well established in the human body. As IM has slower digestion and cleavage rate than SUC, its degradation to GLU and FRU is very slow than sucrose. Resultantly, the blood sugar level remains in the control and does not fluctuate rapidly [26]. The physiological study of IM also showed that strong bonding between two monosaccharides creates different properties of IM from SUC, The difference in physiological properties make IM some benefits over SUC as it contains melting point than SUC, it is highly stable with lower glycemic index [28]. The stability and magnification of transgene expression is a huge challenge in plant biotechnology and genetic engineering. Many efforts in the past were directed to magnify the expression pattern of transgenes in sugarcane crop. The performance of various gene constructs under single, double and triple co-transformed systems evaluated, second most important aspect of the current work. The production of recombinant proteins harvested in sugarcane stems was made possible through the stacking of different constructs transformation leading to the production of sucrose isomerase in sugarcane stems [33]. The *Zea mays* derived ubiquitin promoter (Zm-Ubi) is reported to drive utmost high degree of expression in sugarcane leaves while previous studies witnessed stem specific enhanced expression of transgenes driven under *pCmYLCV* [34]. Previous work revealed that regulated transformation in sugarcane is possible with *pCmYLCV* promoter witnessed much higher stable gene transcription than resulted from *Zm-Ubi* and *CaMV35S* driven line. The *pCmYLCV* promoter was reported to express good in both monocots as well as in dicots [34]. Another work witnessed the development of canker resistant sweet orange genes under *pCmYLCV* promoter produced 89% more expression than other promoter system [35]. Similar studies described the integrity of ubiquitin promoter over CaMV35S [23] in transgenic plants related to various plant groups. Transgenic sugarcane plants possessing genes driven specifically under constitutive promoter ubiquitin witnessed transcription of mRNA with 82.35% expression level by *Zm-Ubi* in comparison to 57.68% obtained from CAMV35S. The *pCmYLCV* promoter is reported to increase transgene expression in Arabidopsis, *Nicotiana tobbacum*, Zea mays and Oryza sativa with extra expression enhancement in meristems, callus, and vegetative and reproductive parts of the plants. Similar work also established the effective role of *pCmYLCV* in boosting the expression of Glyoxalase I gene in blackgram, a different work of its nature. The boosted gene expression and development of the salt tolerant transgenic blackgram established the effective role of *pCmYLCV* promoter. These results were in-line with the outcomes obtained from the present studies. In another research

work, two different versions of *pCmYLCV* promoters, *pCmYLCV9.1* and *pCmYLCV4* were subjected to transgene expression analysis of GUS genes. The work established 28 times higher expression of reporter gene, driven by *pCmYLCV* than expressed under CAMV35S [36] and showed increased expression levels in leaves than in other stem tissues. Many other investigations judged the performance of Polyubiquitine as a promoter in triggering transgene expression pattern especially, in monocots. Further studies also investigated the performance of different variants of ubiquitin, Ubi4 and Ubi9 in both maize and sugarcane plants. After transformation of GUS genes, its expression was investigated in both sugarcane and *Zea mays* in which the results indicated that ubiquitin showed elevated expression level of GUS gene in sugarcane leaves than in the maize plant [37]. Studies also revealed that maize adopted ubiquitin showed lower expression due to post transcriptional gene silencing (PTGS) which was not observed and reported in case of sugarcane plant. Furthermore, the comparative performance of Ubi and CAMV35S promoters were also investigated in Chrysanthemum plants [38]. The results revealed that the activity of Ubi was more durable, increased, and efficient in leaves than CAMV35S and actin promoters. Transgene expressed vigorously in all parts of the chrysanthemum with optimum results of stable transformation. It leads to practical investigations, validating efficacy and commitment of Ubi promoter than the CaMV35S promoter. The expression levels of GTG and Cry1Ac were quantified [25]. Ubiquitin-driven transgenic lines showed much higher expression of transgenes in leaves than CaMV35S-driven transgenic lines [26]. Augmented expression levels of transgenes under Ubiquitin upheld various characteristics entangled with several cis-elements in constitutive promoters. It was attributed that cis elements work under *ImSyGII* aggregated around TSS [27]. Contrastingly compact promoters mostly uncovered the reception of such cis regulators. This research work also employed stem-specific promoter regions derived the Cestrum yellow leaf curl virus. Transgenic lines driven under the *pCmYLCV* promoter seem to be highly operational in sugarcane stems. This promoter host belongs to the C*aulimoviridae* family. The *pCmYLCV* promoter outsourced increased transgene expression in both vegetative and reproductive parts including meristems, but it showed the highest expression in sugarcane stems than leaves. Similar studies produced transgenic lines driven under *pCmYLCV* in various other crops *Arabidopsis thaliana*, *Oryza sativa*, *Nicotiana tobacum* and *Lycopersicon esculentum*. Expression patterns in these crops validated increased expression levels in all parts of the plant in contrast to our studies witnessed elevated transgene activity under *pCmYLCV*. Another interesting feature of *pCmYLCV* promoter is its ability to express maximally in all categories of plants. It performed equally well, and exhibited increased gene expression in monocotyledons as well as in dicotyledons [38, 39]. Another contrasting feature which is irrelevant to the present outcome, is *pCmYLCV* activity in endosperms and pollens, which is not seen in our research work. Further similar investigations evaluated the activity of *pCmYLCV* promoter by expressing multiple genes, related to salinity stress tolerance in sugarcane. Such actions revealed that influential *pCmYLCV* promoter causes increased expression in all sugarcane tissues, not limited to stem tissues and over-expressed salinity tolerant genes [29, 40]. The *pCmYLCV* promoter proved to be effective both for transient as well as stable transformation [30–32]. The execution of transformation by combining three different constructs in the same plant upheld the conceptual approach validating the fact that there is no bound to combinatorial co-transformation events. Many expression vectors can be transformed in sugarcane callus in a single event. The present study adopted a different approach of introducing one gene i.e., *ImSyGII* in sugarcane driven under *Zm-Ubi*, *pCmYLCV*, both in one construct (DPTE). In the last approach, all three gene constructs were co-transformed in a stacked way to ensure an increased gene expression level. Our work demonstrated that transgene indicated the highest expression level when all three constructs are transformed in a combinatorial manner. The data of real-time PCR witnessed

the highest gene expression via stacked co-transformation while, comparatively lowest gene expression by the STE1 construct. A similar approach was also adopted in a work that harvested recombinant protein by using biological systems under various promoters and constructs. The application of the strategy indicated augmented bovine lysozyme protein harvests, originated from sugarcane stems [33]. Hence, the present research work witnessed the enhancement of sugar contents of sugarcane by the induction of *ImSyGII* gene in sugarcane, as compared to control non-transgenic line of the same cultivar. It also established that more than one promoter can help in improving the transgene expression by stable transformation procedure. Furthermore, the authenticity and importance of stem specific constitutive promoter was also came to light after this scientific endeavor. The present study will pave the way for future perspectives of the sugarcane biotechnology so that sugar contents can be enhanced by utilizing the solid biological approaches.

## Supporting information

**S1 File.**
(DOCX)

**S1 Raw images.**
(DOCX)

## Author Contributions

**Conceptualization:** Mudassar Fareed Awan, Abdul Munim Farooq.

**Data curation:** Mudassar Fareed Awan, Sajed Ali, Muhammad Farhan Sarwar, Muhammad Shafiq, Usman Arif, Qurban Ali, Abdul Munim Farooq.

**Formal analysis:** Mudassar Fareed Awan, Sajed Ali, Muhammad Farhan Sarwar, Muhammad Shafiq, Usman Arif, Qurban Ali, Abdul Munim Farooq.

**Investigation:** Qurban Ali, Idrees Ahmad Nasir.

**Methodology:** Sajed Ali, Muhammad Farhan Sarwar, Muhammad Shafiq, Usman Arif.

**Project administration:** Idrees Ahmad Nasir.

**Resources:** Mudassar Fareed Awan, Sajed Ali, Muhammad Farhan Sarwar, Muhammad Shafiq, Usman Arif, Abdul Munim Farooq, Shiming Han.

**Software:** Muhammad Shafiq, Usman Arif, Abdul Munim Farooq, Shiming Han.

**Supervision:** Idrees Ahmad Nasir.

**Validation:** Abdul Munim Farooq, Shiming Han, Idrees Ahmad Nasir.

**Visualization:** Shiming Han, Idrees Ahmad Nasir.

**Writing – original draft:** Mudassar Fareed Awan.

**Writing – review & editing:** Qurban Ali, Shiming Han.

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
