## [Decision Letter · Decision Letter 0]

6 Nov 2023

PONE-D-23-31445

Over expression of Modified Isomaltulose Synthase Gene II (ImSyGII) Under Single and Double Promoters Drive Unprecedented Sugar Contents in Sugarcane

PLOS ONE

Dear Dr. Ali,

Thank you for submitting your manuscript to PLOS ONE. After careful consideration, we have decided that your manuscript does not meet our criteria for publication and must therefore be rejected.

Specifically, the manuscript need substantial improvements in overall language & grammar, methodology especially cloning and transformation part as well as the results section.The results need to be revamped.

I am sorry that we cannot be more positive on this occasion, but hope that you appreciate the reasons for this decision.

Kind regards,

Faiz Ahmad Joyia, Ph.D.

Academic Editor

PLOS ONE

**Additional Editor Comments: **

Over expression of Modified Isomaltulose Synthase Gene II (ImSyGII) Under Single and Double Promoters Drive Unprecedented Sugar Contents in Sugarcane

This research topic is very important for an agricultural country whose sugar production is majorly relying on sugarcane.

• Abstract: Language needs improvement. It’s not very understandable in the first read.

• Which promoter gene combinations were introduced. Either different leaf and stalk specific promoters were used or only one promoter?

• The way author mentioned the cloning and transformation strategy is very ambiguous.

• Which sugarcane genotype was transformed?

• Headings pattern is not uniform throughout the manuscript

• What was the original source of ImSyGII gene?

• Graphical representation of vectors should be improved. Mention the size of each vector.

• In the description of genetic maps, sense of explanation is destroyed in the effort to remove plagiarism.

• Methodology is very poorly written.

• sodium salt 1M NaCl? Recheck your protocol.

• What is the purpose of 6x blue dye staining in coating the gold particles.

• Author should recheck the protocols and their explanation in the manuscript.

Reviewers' comments:

Reviewer's Responses to Questions

**Comments to the Author**

1. Is the manuscript technically sound, and do the data support the conclusions?

Reviewer #1: Yes

Reviewer #2: No

2. Has the statistical analysis been performed appropriately and rigorously? 

Reviewer #1: Yes

Reviewer #2: No

3. Have the authors made all data underlying the findings in their manuscript fully available?

Reviewer #1: Yes

Reviewer #2: Yes

4. Is the manuscript presented in an intelligible fashion and written in standard English?

Reviewer #1: Yes

Reviewer #2: No

5. Review Comments to the Author

Reviewer #1: The manuscript reports expression of isomaltulose synthetic genes in sugarcane through a three gene constructs driven under promoters Zea mays ubiquitin and cestrum yellow leaf curl virus in the single and double combined stacked system. The authors report that employment of three gene constructs collectively produced elite sugar lines producing more than 78% enhancements in whole sugar recovery percentage. They also report that mature internode is highly efficient and receptive for the production of isomaltulose. Overall, they found an increased Brix/ sugar in the transgenic sugarcane lines than control lines. This is a novel finding with wonderful achievement in scientific research and merits publication.

However the authors have to clarify on the comments indicated in the manuscript.

1. Cane yield is an important economic parameter in transgenic lines , but the manuscript is silent on that. Let the authors clarify.

2. However isomaltulose is beneficial to human consumption and how it is going to be economically useful?

3. Is it possible to use such genes in other crops like sugarbeet to improve sugar contents.

4. What is the recovery percentage of the transgenic lines as against the control line?

5. In your findings the role of stem specific promoters were investigated to enhance the production of sugars in transgenic sugarcane. What is the impact of leaf specific promoters in transgenic sugarcane?

6. What is the impact of your study on other features of plants?

7. Do control lines have endogenous ImSyGII gene?

Reviewer #2: This research topic is very important for an agricultural country whose sugar production is majorly relying on sugarcane.

The whole manuscript should be written again.

Abstract: Language needs improvement. It’s not very understandable in the first read.

Which promoter gene combinations were introduced. Either different leaf and stalk specific promoters were used or only one promoter?

Which sugarcane genotype was transformed? Mention it in the abstract.

Headings pattern is not uniform throughout the manuscript.

What was the original source of ImSyGII gene?

Graphical representation of vectors should be improved. Mention the size of each vector.

In the description of genetic maps, sense of explanation is destroyed in the effort to remove plagiarism.

Methodology is very poorly written.

sodium salt 1M NaCl? Recheck your protocol.

What is the purpose of 6x blue dye staining in coating the gold particles.

Do the statistical analysis and explain the significant results in your graphs.

6. PLOS authors have the option to publish the peer review history of their article (what does this mean?). If published, this will include your full peer review and any attached files.

Do you want your identity to be public for this peer review? For information about this choice, including consent withdrawal, please see our Privacy Policy.

Reviewer #1: No

Reviewer #2: No

- - - - -

---

## [Author Response · Author response to Decision Letter 0]

10 Jan 2024

A Rebuttal Letter

The Answers to the editorial and Reviewer’s Comments

PONE-D-23-31445 

”Over expression of Modified Isomaltulose Synthase Gene II (ImSyGII) Under Single and Double Promoters Drive Unprecedented Sugar Contents in Sugarcane”

PLOS ONE

Answers to the Additional Editor Comments : 

Over expression of Modified Isomaltulose Synthase Gene II (ImSyGII) Under Single and Double Promoters Drive Unprecedented Sugar Contents in Sugarcane

This research topic is very important for an agricultural country whose sugar production is majorly relying on sugarcane.

•1. Abstract: Language needs improvement. It’s not very understandable in the first read.

Answer 1: The abstract is made more understandable and language improvement has been done considerably. 

2. Which promoter gene combinations were introduced. Either different leaf and stalk specific promoters were used or only one promoter?

Answer 2:

It was written in the manuscript (page number 4, line 6) in the materials and method section that two different promoters were used in three different promoter combinations. One was leaf specific while other was stem specific while in the third combination both were combined in a single plant expression vector. 

3. The way author mentioned the cloning and transformation strategy is very ambiguous.

Answer 3:

 The cloning and transformation strategy was described in a very detailed manner. Further clarity was also done after careful revisions.

4. Which sugarcane genotype was transformed?

Answer 4: The page 5 (line no. 1) of materials and method section shows that sugarcane genotype HSF-240 was used for transformation. Similarly, in many subsequent pages the name of sugarcane genotype was mentioned time and again.

5. Headings pattern is not uniform throughout the manuscript

Answer 5: Heading pattern is also made uniformed throughout the manuscript.

6. What was the original source of ImSyGII gene?

Answer 6: The original source of the ImSyGII was bacterial strain Pentoea dispersa as described in the manuscript.

7. Graphical representation of vectors should be improved. Mention the size of each vector.

Answer 7: The size of the constructs with gene fragment sizes are given in the materials and method section and in maps figure no. 1A.B.C respectively.

8. In the description of genetic maps, sense of explanation is destroyed in the effort to remove plagiarism.

Answer 8:

In the description of genetic maps, all figures have been explained in a very detailed way. Figures description does not need any change to avoid plagiarism because already similarity index of the manuscript is very low and figures description have nothing to do with plagiarism.

9. Methodology is very poorly written.

Answer 9: Methodology is improved as suggested. 

10. sodium salt 1M NaCl? Recheck your protocol.

Answer 10: High salt concentration is necessary to avoid any possible degradation moreover to create high alkaline conditions therefore salt is used . The adopted protocol was rechecked.

11. What is the purpose of 6x blue dye staining in coating the gold particles.

Answer 11:

I have not used gold particles while used tungsten particles as described in (page 6 line 2) the submitted manuscript. Secondly, 6x blue dye staining was done to indicate the position of bombarded materials into the sugarcane callus. 

12. Author should recheck the protocols and their explanation in the manuscript.

Answer 12: Authors have checked the protocols and their explanation was made in a very described way in the manuscript. Moreover, in case of any specific indication it can be changed accordingly. 

5. Review Comments to the Author

Comments of Reviewer #1 and Answers: 

The manuscript reports expression of isomaltulose synthetic genes in sugarcane through a three gene constructs driven under promoters Zea mays ubiquitin and cestrum yellow leaf curl virus in the single and double combined stacked system. The authors report that employment of three gene constructs collectively produced elite sugar lines producing more than 78% enhancements in whole sugar recovery percentage. They also report that mature internode is highly efficient and receptive for the production of isomaltulose. Overall, they found an increased Brix/ sugar in the transgenic sugarcane lines than control lines. This is a novel finding with wonderful achievement in scientific research and merits publication.

However the authors have to clarify on the comments indicated in the manuscript.

1. Cane yield is an important economic parameter in transgenic lines , but the manuscript is silent on that. Let the authors clarify.

Answer 1: The present work has used the scientific strategy to enhance the sugar recovery percentage of the sugarcane which will ultimately enhance the sugar yield.

2. However isomaltulose is beneficial to human consumption and how it is going to be economically useful?

Answer 2: Isomaltulose can be extracted, purified and can be used as an artificial sweetener in our industries like jam jellies and other food products.

3. Is it possible to use such genes in other crops like sugarbeet to improve sugar contents.

Answer 3: Yes, it is possible to use these genes in other food crops like sugarbeet. Even, we can enhance the sugar contents of fruits by employing these genes.

4. What is the recovery percentage of the transgenic lines as against the control line?

Answer 4: The transgenic line in our study produced 21% sugar recovery as in TCTE lines than control (10%).

5. In your findings the role of stem specific promoters were investigated to enhance the production of sugars in transgenic sugarcane. What is the impact of leaf specific promoters in transgenic sugarcane?

Answer 5: The leaf specific promoters may enhance the vegetative growth and can show good gene product at leaves which are not required in the present study while stem specific promoters will enhance the gene activity at stems conclusively adding other phenotypic features in transgenic stems as described in our study.

6. What is the impact of your study on other features of plants?

Answer 6: The plants may have high photosynthetic efficiency and can produce better sugar signaling mechanism than control lines.

7. Do control lines have endogenous ImSyGII gene?

Answer 7: No, the control lines don’t have any ImSyGII gene.

Answers to the reviewer 2 comments

Reviewer #2: This research topic is very important for an agricultural country whose sugar production is majorly relying on sugarcane.

1. The whole manuscript should be written again.

Answer 1: The whole manuscript is proofread and improved accordingly with necessary corrections.

2. Abstract: Language needs improvement. It’s not very understandable in the first read.

Answer 2: The language of the abstract is made clear and understandable. It is written in a very scientific manner. 

3. Which promoter gene combinations were introduced. Either different leaf and stalk specific promoters were used or only one promoter?

Answer 3:

It was written very clearly in my manuscript (page number 4, line 6) in the materials and method section that two different promoters were used in three different combinations. One was leaf specific while other was stem specific while in the third combination both were combined in a single plant expression vector. The STE1 was used for expression in leaves while STE2 was used for expression of ImSyGII in stem.

4. Which sugarcane genotype was transformed? Mention it in the abstract.

Answer 4: The page 5 (line no. 1) of materials and method section shows that sugarcane genotype HSF-240 was used for transformation. Similarly, in many subsequent pages the name of sugarcane genotype was mentioned time and again. I will add it in the abstract section.

5. Headings pattern is not uniform throughout the manuscript.

Answer 5: Heading pattern is made uniform throughout the manuscript and if there is any ambiguity, please mention clearly. I will change it accordingly.

6. What was the original source of ImSyGII gene?

Answer 6: The original source of the ImSyGII was bacterial strain Pentoea dispersa as described.

7. Graphical representation of vectors should be improved. Mention the size of each vector.In the description of genetic maps, sense of explanation is destroyed in the effort to remove plagiarism.

Answer 7:

The graphical representation of the vectors can be improved as advised. In the description of genetic maps, all figures have been explained in a very detailed way. If editor needs any specific details it can be added. Figures description does not need any change to avoid plagiarism because already similarity index of the manuscript is very low and figures description have nothing to do with plagiarism.

Answer 8: The size of the constructs with gene fragment sizes are given in the materials and method section and in maps figure no. 1A.B.C respectively. 

9. Methodology is very poorly written.

Answer 9: Methodology was written in a very detailed way without any grammatical errors. However, it is further Improved. If you find any error, it must be mentioned specifically. 

10. sodium salt 1M NaCl? Recheck your protocol.

Answer 10: Dear reviewer 2, high salt concentration is necessary to avoid any possible degradation moreover to create high alkaline conditions therefore salt is used . The adopted protocol was rechecked.

11. What is the purpose of 6x blue dye staining in coating the gold particles.

Answer 11: I have not used gold particles while used tungsten particles as described in (page 6 line 2) the submitted manuscript. Secondly, 6x blue dye staining was done to indicate the position of bombarded materials into the sugarcane callus. 

12. Do the statistical analysis and explain the significant results in your graphs.

Answer 12: All the figures mentioned and graphs showed error bars show respective standard deviation of any result. If you want any specific statistical analysis please do mention I have already provided further statistical analysis as supplementary file.

---

## [Decision Letter · Decision Letter 1]

24 Jun 2024

PONE-D-23-31445R1Over expression of Modified Isomaltulose Synthase Gene II (ImSyGII) Under Single and Double Promoters Drive Unprecedented Sugar Contents in SugarcanePLOS ONE

Dear Dr. Ali,

Thank you for submitting your manuscript to PLOS ONE. After careful consideration, we feel that it has merit but does not fully meet PLOS ONE’s publication criteria as it currently stands. Therefore, we invite you to submit a revised version of the manuscript that addresses the points raised during the review process.

We have now received four completed reviews from additional reviewers after the original reviewers were not available to re-review; the comments are available below. The additional reviewers have raised significant remaining scientific concerns about the study that need to be addressed in a revision.

Please revise the manuscript to address all the reviewer's comments in a point-by-point response in order to ensure it is meeting the journal's publication criteria. Please note that the revised manuscript will need to undergo further review, we thus cannot at this point anticipate the outcome of the evaluation process.

We look forward to receiving your revised manuscript.

Kind regards,

Miquel Vall-llosera Camps

Senior Staff Editor

PLOS ONE

Journal Requirements:

a) The name of the colleague or the details of the professional service that edited your manuscript.

b) A copy of your manuscript showing your changes by either highlighting them or using track changes (uploaded as a *supporting information* file).

c) A clean copy of the edited manuscript (uploaded as the new *manuscript* file).

3. We note that you have provided funding information that is not currently declared in your Funding Statement. However, funding information should not appear in the Acknowledgments section or other areas of your manuscript. We will only publish funding information present in the Funding Statement section of the online submission form. Please remove any funding-related text from the manuscript.

7. Please upload a copy of Figure 13, to which you refer in your text on page 35 (in PDF format). If the figure is no longer to be included as part of the submission please remove all reference to it within the text.

Reviewers' comments:

Reviewer's Responses to Questions

**Comments to the Author**

1. If the authors have adequately addressed your comments raised in a previous round of review and you feel that this manuscript is now acceptable for publication, you may indicate that here to bypass the “Comments to the Author” section, enter your conflict of interest statement in the “Confidential to Editor” section, and submit your "Accept" recommendation.

Reviewer #3: All comments have been addressed

Reviewer #4: (No Response)

Reviewer #5: (No Response)

Reviewer #6: (No Response)

2. Is the manuscript technically sound, and do the data support the conclusions?

Reviewer #3: Yes

Reviewer #4: Partly

Reviewer #5: Yes

Reviewer #6: Yes

3. Has the statistical analysis been performed appropriately and rigorously? 

Reviewer #3: Yes

Reviewer #4: I Don't Know

Reviewer #5: Yes

Reviewer #6: Yes

4. Have the authors made all data underlying the findings in their manuscript fully available?

Reviewer #3: Yes

Reviewer #4: Yes

Reviewer #5: Yes

Reviewer #6: Yes

5. Is the manuscript presented in an intelligible fashion and written in standard English?

Reviewer #3: Yes

Reviewer #4: No

Reviewer #5: Yes

Reviewer #6: Yes

6. Review Comments to the Author

Reviewer #3: Dear Authors

Many thanks for revising the manuscript by considering the comments/suggestions given by Reviewers.

Regards

Reviewer #4: Production of isomaltoluse (IM) that more healthy and sweeter with low lower glycemic index than sucrose is important research for the industry. This manuscript was directed to enhance the content of IM in transgenic sugarcane. The strategy to increase IM was overexpression of isomaltulose synthase gene II (ImSyGII) under single, double, and triple promoter. The authors claimed that the employment of three gene constructs produced elite sugarcane transgenic lines. However, the authors have to clarify bellow comments.

1. It is not clear the reason behind to use single, double, and triple promoter, because the promoter that used in the experiment both Zm-pUbi and pCmYLCV are constitutive promoters and should be expressed in all tissue. But the results showed that Zm-pUbi efficiently expressed in leave and pCmYLCV in stalk.

2. Fig. 1 Illustration of the genetic map should be completed with restriction sites. The figure should also explain whether the Zm-pUbi or pCmYLCV located in the upstream or downstream in the double promoter construct. There is GUS reported gene in the construct, why the experiment did not employ the gene to decide localization of gene expression.

3. Method for screening of putative transgenic was difficult to understand. How many incubations (subculture) in the selection media? Is that true only two times? And how many percentages of transformation efficiency were not presented. Furthermore, how to differentiate the putative transformant that containing triple stacked construct during screening, since all DNA constructs containing same antibiotic resistance, hygromycin. What is the control plant in the last sentence of 2.3. section.

4. PCR amplification of IMSyGII gene was used for screening of transgenic plants, but there was no figure presenting the electrophoresis of the PCR product. Is that true that gel electrophoresis was performed to separate amplicons using 2.5% agarose instead of 1%?

5. Section 2.5. Estimation of copy number by real time PCR. What is the purpose of this section, to identify of the DNA copy number instead of Southern Blot analysis? If the section was proposed to identify the gene copy number, the Southern Blot analysis have to be conducted. However, if the section was directed to analysis the gene expression levels using RT-PCR, the experiment should use RNA. There is no sentence explaining RNA isolation method. In addition, there was no nucleotide primers sequence for qRT-PCR analysis of ImSYGTII gene and beta-actin gene presented in this paper.

6. Section 3.1. (Results), Fig 2 (ABCD) should be completed with the used restriction enzymes. The figures (2ABC) were also unclear, the digestion resulted a clear 2 DNA bands, but there was no DNA band for the rest DNA vector after digestion?

7. Fig 3 and 4, they should be removed from the paper because only presenting the survive callus and plantlet. What about the died materials? This is important data in order to calculate transformation efficiency.

8. Figure 6, Why the transgenic plants were cultivated in open environment? The transgenic plants are prohibited to cultivated in the oven environment (field) before biosafety certifications for community acceptance. Fig 6 was not presented properly as the experiment, and just presented as cropping image. In addition, the authors should consider the effect of environment on the gene expression since the plants were grown in oven environment.

9. Fig. 7, 8, 9 10, what is the unit of expression levels? And please be consistent in using the words stem, stalk or culm.

10. Fig 7 – 10 were the expression levels and Fig 12 is distribution of sugar contents. The expression was increased as well as IM contents, but there no prove on isomaltulose synthase activity? It is better to include the activity in the paper since the increase in gene expression levels are not always accompanied by the increase of the metabolite.

11. The manuscript mentioned Fig 13, but in fact the Fig 13 is not found.

12. The English is difficult to understand, need revise in the English editing services. There are some mistyping in the manuscripts that should be revised.

Reviewer #5: Comments for Reviewer:

The research work seems to be interesting and valuable and conducted in a very detailed manner. The authors have mentioned the protocols and material and methods in a detail. The results were interpreted nicely. However, to enhance the clarity and readability of the manuscript, respond the following questions:

1. Add one to two introductory lines in abstract.

2. Is Sucrose isomerase present endogenously in sugarcane?

3. How the screening of putative transgenic plants were made at initial level?

4. Can this approach be applicable in other food or sugar crops?

5. What size of transgene (ImSyGII) was cloned in plant expression vector.

6. Which sugarcane cultivar was used as a control plant?

7. Is there any chance of PTGS in sugarcane against this transgene when it integrated in sugarcane genome?

8. At which age the callus of sugarcane was in highly receptive stage of transformation.

9. The direct method of transformation was used in this research work, however some of the researchers have also used agrobacterium mediated transformation for monocots do you think that gene gun methods is still a preferred method for monocots?

10. What was the annealing temperature of PCR primers used in this study?

11. What is the way forward of your research work.

Reviewer #6: 1. Is the enzyme sucrose isomerase present endogenously in sugarcane?

2. Which sugarcane cultivar was used for calli generation and which particles were used for DNA adsorption.

3. Your work seems impressive and promising in enhancing sugar contents in sugarcane as described, is there any possibility, the same genes can be introduced in other food crops?

4. How the quantifications of sugar contents was obtained in this work?

5. The authors exploited three different promoter combinations in this work by using sugar enhancing genes and observed different gene products at different tissues. Which promoter combination is optimum in producing maximum sugar enhancements in sugarcane culms?

6. Is there any chance of PTGS in sugarcane against these genes

7. PLOS authors have the option to publish the peer review history of their article (what does this mean?). If published, this will include your full peer review and any attached files.

Reviewer #3: **Yes: **Muthukrishnan Arun

Reviewer #4: No

Reviewer #5: No

Reviewer #6: No

---

## [Author Response · Author response to Decision Letter 1]

29 Aug 2024

Answers to the Reviewers

Comments and Answers to Reviewer 4

Reviewer #4: Production of isomaltulose (IM) that more healthy and sweeter with low lower glycemic index than sucrose is important research for the industry. This manuscript was directed to enhance the content of IM in transgenic sugarcane. The strategy to increase IM was overexpression of isomaltulose synthase gene II (ImSyGII) under single, double, and triple promoter. The authors claimed that the employment of three gene constructs produced elite sugarcane transgenic lines. However, the authors have to clarify bellow comments.

Comment 1: It is not clear the reason behind to use single, double, and triple promoter, because the promoter that used in the experiment both Zm-pUbi and pCmYLCV are constitutive promoters and should be expressed in all tissue. But the results showed that Zm-pUbi efficiently expressed in leave and pCmYLCV in stalk.

Answer 1: In fact different promoter combinations were used to evaluate their performance to produce maximum gene expression in different tissues. Although both promoters are constitutive but their performance varies from tissue to tissue depending upon various factors like the presence of enhancer sequences, turning on and off of various proteins, abundance of transcriptional machinery and their evolutionary adaptations. Furthermore, all of these transformation events are independent therefore the expression results may vary. Moreover, it is not necessary that all constitutive promoters perform equally in all tissues, there are many tissue specific promoters perform differently on different conditions described above. 

2. Fig. 1 Illustration of the genetic map should be completed with restriction sites. The figure should also explain whether the Zm-pUbi or pCmYLCV located in the upstream or downstream in the double promoter construct. There is GUS reported gene in the construct, why the experiment did not employ the gene to decide localization of gene expression.

Answer 2:

I am very thankful to the reviewer for his valuable suggestion. The restriction sites were added in the genetic map. The other details regarding downstream or upstream location of promoters are also added in the figure description as per your directions.Yes you are absolutely right that the GUS expression will definitely help in localization of transgene. Our point is that we could not add GUS because our focus was totally into the enhancement of sugar recovery percentage in addition to evaluation of using single, double or stacked promoters. Secondly, we have very limited resources and we need to manage accordingly.

3. Method for screening of putative transgenic was difficult to understand. How many incubations (subculture) in the selection media? Is that true only two times? And how many percentages of transformation efficiency were not presented. Furthermore, how to differentiate the putative transformant that containing triple stacked construct during screening, since all DNA constructs containing same antibiotic resistance, hygromycin. What is the control plant in the last sentence of 2.3. section.

Answer 3:

The putative transgenic sugarcane plants were screened through selection media. As sugarcane is a fast growing plant therefore only two sub-cultures of selection media were performed which were enough to screen positive putative transgenic plants from non-transgenic plants. We have missed to add transformation efficiency and we are very thankful for honorable reviewer to point out it. We have added it in the manuscript. Moreover, the initial screening of all putative transgenic plants with single, double and triple stacked constructs were made only on the basis of hygromycin as all contain the similar vector with only small difference of promoters. The further differentiation of single, double and triple stacked promoters were already done by proper labeling and PCR analysis. The control plants in section 2.3 were shown in figure 4. Some of the control plants with no transgene could not survive in test tubes, figure 3. The control Plants belong to variety HSF-240 which was not transformed with ImSyGII.

4. PCR amplification of IMSyGII gene was used for screening of transgenic plants, but there was no figure presenting the electrophoresis of the PCR product. Is that true that gel electrophoresis was performed to separate amplicons using 2.5% agarose instead of 1%?

Answer 4:

The PCR amplifications of ImSyGII was made clear in the figure 2 (D) indicating the 500bp of PCR product. The PCR gel pic showing proper 500bp band was the result of gel electrophoresis with agarose gel (2.5%). The 2.5% agarose gel provides much better resolution for small DNA fragments moreover, the gel was available and I did not want to waste resources thatswhy I used this 2.5% gel. As the product size in our gene was small fragment therefore 2.5% agarose gel was used. The increased concentration of agarose creates a denser matrix, slowing down the migration of smaller DNA fragments and making them to be separated more distinctly.

5. Section 2.5. Estimation of copy number by real time PCR. What is the purpose of this section, to identify of the DNA copy number instead of Southern Blot analysis? If the section was proposed to identify the gene copy number, the Southern Blot analysis have to be conducted. However, if the section was directed to analysis the gene expression levels using RT-PCR, the experiment should use RNA. There is no sentence explaining RNA isolation method. In addition, there was no nucleotide primers sequence for qRT-PCR analysis of ImSYGTII gene and beta-actin gene presented in this paper.

Answer 5: 

The purpose of this section is to show the expression analysis of positive transgenic sugarcane plants through RT-PCR. The RNA extraction protocol with necessary primer information has been added in this section 2.5. It has been made more clear after your valuable suggestion. The copy number estimation was not the part of the study in this case and hence removed from the paragraph.

6. Section 3.1. (Results), Fig 2 (ABCD) should be completed with the used restriction enzymes. The figures (2ABC) were also unclear, the digestion resulted a clear 2 DNA bands, but there was no DNA band for the rest DNA vector after digestion?

Answer 6:

The used restriction enzymes and their details are added in the figure description as suggested. The two bands which were resulted in figure 2 ABC showed confirmation of one promoter and other was the remaining DNA vector as shown in figure 2. The figure 2A shows the separation of ubiquitin promoter from the rest of the vector as confirmation while other band shows the vector, fig 2B similarly shows the digestion of ImSyGII while upper band (4790bp) indicated the cut cloning DNA vector, while in fig. 2C both promoters were cut for confirmation while the upper band showed the remaining vector DNA.

7. Fig 3 and 4, they should be removed from the paper because only presenting the survive callus and plantlet. What about the died materials? This is important data in order to calculate transformation efficiency.

Answer 7: 

The figures 3 and 4 showed the process of callus induction and their regeneration after particle gene bombardment purposed. If reviewer 4 thinks its unnecessary to show in this condition, I will remove it from the manuscript after your direction. Figures 3 and 4 have been removed accordingly.

8. Figure 6, Why the transgenic plants were cultivated in open environment? The transgenic plants are prohibited to cultivated in the oven environment (field) before biosafety certifications for community acceptance. Fig 6 was not presented properly as the experiment, and just presented as cropping image. In addition, the authors should consider the effect of environment on the gene expression since the plants were grown in oven environment.

Answer 8:

The transgenic sugarcane plants were initially shifted to plastic pots in glass house conditions which were designated for GMO restricted experimental areas. Moreover, these Sugarcane GMOs were not cultivated publicly in open areas but the figure shows the experimental fields which were designated restricted area for experimental plants in National Centre of Excellence in Molecular Biology. Moreover, in lahore the sugarcane plant does not flower due to its climatic conditions. These plants are now being assessed for their biosafety studies and will be cultivated openly after the approval of national certifications. The figure 6 shows the cultivation in experimentation areas to estimate their sugar recovery percentage after allowing them to attain full maturity. Moreover, the effect of environment in gene expression and sugar contents was a very large topic which will be discussed in our another manuscript “Field performance and the activity of ImSyGII in successive VG1 and VG2 of transgenic sugarcane Lines”. The data in this paper was already very large therefore we adopted to cover it another paper.

9. Fig. 7, 8, 9 10, what is the unit of expression levels? And please be consistent in using the words stem, stalk or culm.

Answer 9:

The expression level in RT-PCR does not have any unit. It was based on relative term and was always shown with respect to the control. We always represent expression relative to control plant which was statistically converted to 1. All other values of transgenic plants were shown in for example 2times, 3 times like that as shown in figures 7, 8,9 and 10. The words stem, stalk and culm are the synonym of each other. I will make it consistent for the understanding of readers as per your suggestion.

10. Fig 7 – 10 were the expression levels and Fig 12 is distribution of sugar contents. The expression was increased as well as IM contents, but there no prove on isomaltulose synthase activity? It is better to include the activity in the paper since the increase in gene expression levels are not always accompanied by the increase of the metabolite.

Answer 10:

Respected Sir, I have performed this work in two stages, the first stage represents the molecular work with necessary genotypic and phenotypic data as shown in the paper. The second work was the metabolic study of ImSyGII which was also done but this whole data was kept in another complete study. The metabolic data work totally different enzymatic work which was written in a separate manuscript which will soon be submitted in the same journal. If I would add the whole data in one manuscript it will become very huge thatswhy I divided it into two halves molecular and metabolic studies respectively.

11. The manuscript mentioned Fig 13, but in fact the Fig 13 is not found.

Answer 11:

It was a typing error which is removed.

12. The English is difficult to understand, need revise in the English editing services. There are some mistyping in the manuscripts that should be revised.

Answer 12:

The language is made now easy, understandable and the whole manuscript is revised to improve further.

Answers to the Comments of Reviewer 5

Reviewer #5: Comments for Reviewer:

The research work seems to be interesting and valuable and conducted in a very detailed manner. The authors have mentioned the protocols and material and methods in a detail. The results were interpreted nicely. However, to enhance the clarity and readability of the manuscript, respond the following questions:

1. Add one to two introductory lines in abstract.

Answers 1: Done

2. Is Sucrose isomerase present endogenously in sugarcane?

Answer 2:

No, The sucrose isomerase is not present endogenously in sugarcane. It is added by my study.

3. How the screening of putative transgenic plants were made at initial level?

Answer 3:

The initial screening of putative transgenic plants were made by antibiotic resistant selection media as described in section 2.1.

4. Can this approach be applicable in other food or sugar crops?

Answer 4: 

Yes, It is highly recommended by our work that this scientific approach can be adopted in other food crops also especially in fruits and sugar crops like sugarbeet and stevia.

5. What size of transgene (ImSyGII) was cloned in plant expression vector.

Answer 5:

The transgene (ImSyGII) having 1899bp was cloned in expression vector.

6. Which sugarcane cultivar was used as a control plant?

Answer 6: The sugarcane cultivar HSF-240 was used in this study.

7. Is there any chance of PTGS in sugarcane against this transgene when it integrated in sugarcane genome?

Answer 7: This gene ImSyGII has very good ability to resist PTGS in sugarcane as already reported in previous studies. My another study will also focus on it.

8. At which age the callus of sugarcane was in highly receptive stage of transformation.

Answer 8: The best stage for transformation of transgene in callus is 28-30 days. The callus of this age is considered optimal for transformation.

9. The direct method of transformation was used in this research work, however some of the researchers have also used agrobacterium mediated transformation for monocots do you think that gene gun methods is still a preferred method for monocots?

Answer 9: In fact, monocots can use both methods for transformation but direct gene transfer method by gene gun is considered much better due to increase transformation efficiency and regeneration ability percentage. 

10. What was the annealing temperature of PCR primers used in this study?

Answer 10: The annealing temperature used for this study was 58°C

11. What is the way forward of your research work.

Answer 11: The way forward in this study is that we must exploit other sugar responsive genes in bacterial species and used to produce Isomaltulose. This sugar alternative is highly nutritious and sweeter than other sugar molecules. Moreover, these genes should also be used in other food crops and fruits to enhance sugar contents which are necessary.

Comments of Reviewer 6

Reviewer #6: 1. Is the enzyme sucrose isomerase present endogenously in sugarcane?

Answer 1: No, The sucrose isomerase is not present endogenously in sugarcane. It is added by my study.

2. Which sugarcane cultivar was used for calli generation and which particles were used for DNA adsorption.

Answer 2:

There were many sugarcane cultivars which were evaluated for their ability to produce callus. Their callus induction and regeneration percentages were studied. After study it was seen that HSF-240 presented the optimum ability to produce callus and regeneration.

3. Your work seems impressive and promising in enhancing sugar contents in sugarcane as described, is there any possibility, the same genes can be introduced in other food crops?

Answer 3:

Yes, these genes can be transformed in other food crops like sugarbeet, stevia, mango, apple and banana. It will help producing not only sweeter fruits but also highly nutritious ones.

4. How the quantifications of sugar contents was obtained in this work?

Answer 4: Two different strategies were utilized in this study to obtain quantified sugar concentrations. In first strategy the whole sugar contents and sugar recovery percentage was tested by Brix method which is a commercial method used by sugar mills. More specifically, the Isomaltulose contents were quantified by HPLC. The data of both strategies were given in the manuscripts in the form of figure9,10,11 and 12.

5. The authors exploited three different promoter combinations in this work by using sugar enhancing genes and observed different gene products at different tissues. Which promoter combination is optimum in producing maximum sugar enhancements in sugarcane culms?

Answer 5: The study revealed that if we use ubiquitin promoter in combination with CmYLCV promoter designated as Double promoters in this manuscript, They will produce more expression of the gene than single promoter. The expression will be more in stems than leaves. 

6. Is there any chance of PTGS in sugarcane against these genes?

Answer 6:

The PTGS is very common in sugarcane genome due to very complexity and hugeness. But our studies revealed that there is only single miRNA present in sugarcane which can target ImSyGII. Hence, the chances of PTG

---

## [Decision Letter · Decision Letter 2]

25 Sep 2024

Over expression of Modified Isomaltulose Synthase Gene II (ImSyGII) Under Single and Double Promoters Drive Unprecedented Sugar Contents in Sugarcane

PONE-D-23-31445R2

Dear Dr. Ali,

We’re pleased to inform you that your manuscript has been judged scientifically suitable for publication and will be formally accepted for publication once it meets all outstanding technical requirements.

Kind regards,

S.V. Ramesh, PhD

Academic Editor

PLOS ONE

Additional Editor Comments (optional):

Reviewers' comments:

Reviewer's Responses to Questions

**Comments to the Author**

1. If the authors have adequately addressed your comments raised in a previous round of review and you feel that this manuscript is now acceptable for publication, you may indicate that here to bypass the “Comments to the Author” section, enter your conflict of interest statement in the “Confidential to Editor” section, and submit your "Accept" recommendation.

Reviewer #3: All comments have been addressed

Reviewer #5: All comments have been addressed

2. Is the manuscript technically sound, and do the data support the conclusions?

Reviewer #3: Yes

Reviewer #5: Yes

3. Has the statistical analysis been performed appropriately and rigorously? 

Reviewer #3: Yes

Reviewer #5: Yes

4. Have the authors made all data underlying the findings in their manuscript fully available?

Reviewer #3: Yes

Reviewer #5: Yes

5. Is the manuscript presented in an intelligible fashion and written in standard English?

Reviewer #3: Yes

Reviewer #5: Yes

6. Review Comments to the Author

Reviewer #3: (No Response)

Reviewer #5: (No Response)

7. PLOS authors have the option to publish the peer review history of their article (what does this mean?). If published, this will include your full peer review and any attached files.

Reviewer #3: **Yes: **Muthukrishnan Arun

Reviewer #5: **Yes: **Adnan Iqbal

---

## [Editor Report · Acceptance letter]

8 Oct 2024

PONE-D-23-31445R2 

PLOS ONE

Dear Dr. Ali, 

I'm pleased to inform you that your manuscript has been deemed suitable for publication in PLOS ONE. Congratulations! Your manuscript is now being handed over to our production team.

Kind regards, 

on behalf of

Dr. Shunmugiah Veluchamy Ramesh 

Academic Editor

PLOS ONE